# Multi-objective Differentiable Neural Architecture Search

**Rhea Sanjay Sukthanker** [* 1]   **Arber Zela** [* 1]   **Benedikt Staffler** [2]   **Samuel Dooley** [3]
**Josif Grabocka** [4]   **Frank Hutter** [5 1]

## Abstract

Pareto front profiling in multi-objective optimization (MOO), i.e. finding a diverse set of Pareto optimal solutions, is challenging, especially with expensive objectives like neural network training. Typically, in MOO neural architecture search (NAS), we aim to balance performance and hardware metrics across devices. Prior NAS approaches simplify this task by incorporating hardware constraints into the objective function, but profiling the Pareto front necessitates a computationally expensive search for each constraint. In this work, we propose a novel NAS algorithm that encodes user preferences for the trade-off between performance and hardware metrics, and yields representative and diverse architectures across multiple devices in just one search run. To this end, we parameterize the joint architectural distribution across devices and multiple objectives via a hypernetwork that can be conditioned on hardware features and preference vectors, enabling zero-shot transferability to new devices. Extensive experiments with up to 19 hardware devices and 3 objectives showcase the effectiveness and scalability of our method. Finally, we show that, without extra costs, our method outperforms existing MOO NAS methods across a broad range of qualitatively different search spaces and datasets, including MobileNetV3 on ImageNet-1k, an encoder-decoder transformer space for machine translation and a decoder-only transformer space for language modelling.

## 1. Introduction

The ability to make good tradeoffs between predictive accuracy and efficiency (in terms of latency and/or energy consumption) has become crucial in an age of ever increasing neural networks complexity and size (Kaplan et al., 2020; Hoffmann et al., 2022; Zhai et al., 2022; Alabdulmohsin et al., 2023) and a plethora of embedded devices. However, finding the right trade-off remains a challenging task that typically requires human intervention and a lot of trial-and-error across devices. With multiple conflicting objectives, it becomes infeasible to optimize all of them simultaneously and return a single solution. Ideally, NAS should empower users to choose from a set of diverse Pareto optimal solutions that represent their preferences regarding the trade-off between objectives.

Neural Architecture Search (NAS) (White et al., 2023) provides a principled framework to search for network architectures in an automated fashion. Significant research (Elsken et al., 2019b; Cai et al., 2020; Wang et al., 2020; Chen et al., 2021a) has extended NAS for multi-objective optimization (MOO), considering performance and hardware efficiency metrics like latency and energy consumption. However, to the best of our knowledge, no existing gradient-based method returns the full Pareto front for the MOO problem at hand *without repeating their search routine multiple times with different hardware constraints*.

In this work, we propose a scalable and hardware-aware **M**ulti-**O**bjective **D**ifferentiable **N**eural **A**rchitecture **S**earch (**MODNAS**) algorithm that efficiently trains a single supernet which can be used to read off Pareto-optimal solutions for different user preferences and different target devices, without any extra search steps. To search across devices, we frame the problem at hand as a multi-task multi-objective optimization problem, where each task (device) has multiple (conflicting) objectives, e.g., classification accuracy and latency. The user's preferences are modelled by a *preference vector* that defines a scalarization of the different objectives. This preference vector, along with features of the hardware of interest, is fed to a hypernetwork (Ha et al., 2017) that outputs continuous architectural parameters $\alpha$. To search in the space of architectures, we employ a one-shot model and a bi-level optimization scheme, as is typically done in gradient-based NAS. In our case, however, the upper-level parameters are the hypernetwork weights, optimized in expectation across different preference vectors and hardware devices via multiple gradient descent (Désidéri, 2012).

---

[*]Equal contribution  [1]University of Freiburg [2]BCAI [3]Abacus.AI [4]University of Technology Nuremberg [5]ELLIS Institute Tübingen. Correspondence to:  <sukthank@cs.uni-freiburg.de>.

Accepted to the Workshop on Advancing Neural Network Training at International Conference on Machine Learning (WANT@ICML 2024).

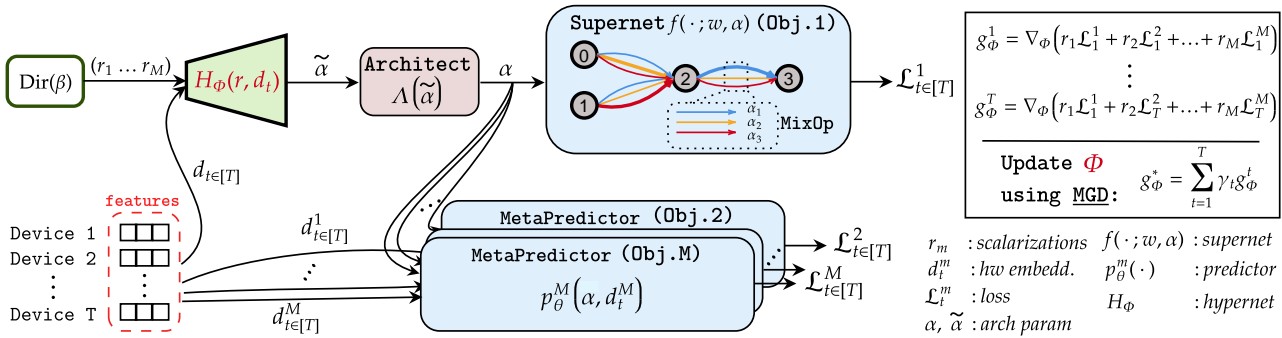

*Figure 1.* **MODNAS overview.** Given a set of $T$ devices, MODNAS seeks to optimize $M$ (potentially conflicting) objectives across these devices. To this end, it employs a `MetaHypernetwork` $H_\Phi(r, d_t)$, that takes as input a scalarization $r$, representing the user preferences, and a device embedding $d_t$, to yield an un-normalized architectural distribution $\tilde{\alpha}$. The `Architect` uses $\tilde{\alpha}$ to sample differentiable discrete architectures, used in the `Supernetwork` to estimate accuracy and in the `MetaPredictor` to estimate the other $M-1$ loss functions (e.g. latency, energy consumption) for every device. By iterating over devices and sampling scalarizations uniformly from the $M$-dimensional simplex, at each iteration we update the `MetaHypernetwork` using multiple gradient descent (MGD).

To evaluate our method, we conduct experiments on multiple NAS search spaces, including CNN and Transformer architectures, and up to 3 objectives across 19 hardware devices. While other NAS methods that utilize hardware constraints in their search objectives require substantial search costs both for each new constraint and each new hardware, MODNAS addresses both in a zero-shot manner, without extra search cost, while nevertheless yielding higher quality solutions. Our contributions can be summarized as follows:

1. We present a principled and robust approach for **Multi-objective Differentiable NAS**, that leverages *hypernetworks* and *multiple gradient descent* to simultaneously find Pareto-optimal architectures across devices.

2. This work is the *first* to provide a global view of the Pareto solutions with **just a single model**, without the need to repeat search or fine-tune on new target devices.

3. Extensive evaluation of our method across **4 different search spaces** (NAS-Bench-201, MobileNetV3, an encoder-decoder and a decoder-only Transformer space), **3 tasks** (image classification, machine translation and language modeling), and **up to 19 hardware devices and 3 objectives**, show both improved *efficiency* and *performance* in comparison to previous approaches that use a constrained objective in their search.

To facilitate reproducibility, we provide our code via the following link.

## 2. Background and Related Work

In this section, before describing our algorithm, we introduce some basic concepts, definitions and related work. Refer to Appendix B for an extended related work.

**Multi-objective optimization (MOO) for Multi-Task Learning.** Consider a multi-task dataset $\mathcal{D}$ consisting of $N$ instances, where the feature vector of the $i-$th instance is denoted as $x_i \in \mathcal{X}$, and the $M$-many associated target variables as $y_i^1 \in \mathcal{Y}^1, \ldots, y_i^M \in \mathcal{Y}^M$. Moreover, consider there exists a family of parametric models $f(\boldsymbol{x}; \boldsymbol{w}) : \mathcal{X} \to \{\mathcal{Y}^1 \times \cdots \times \mathcal{Y}^M\}$, parameterized by $\boldsymbol{w}$, that maps the input $\boldsymbol{x}$ to the joint space of the multiple tasks. To simplify the notation, we denote the prediction of the $m$-th task as $f^m(\boldsymbol{x}; \boldsymbol{w}) : \mathcal{X} \to \mathcal{Y}^m$, and the respective loss $\mathcal{L}^m(\boldsymbol{w}) \triangleq \frac{1}{N}\sum_i^N \ell^m(y_i^m, f^m(x_i; \boldsymbol{w}))$. The vector of the values of all loss functions is denoted as $\mathbf{L}(\boldsymbol{w}) \triangleq (\mathcal{L}^1(\boldsymbol{w}), \ldots, \mathcal{L}^M(\boldsymbol{w}))$. MOO then seeks to find a set of Pareto-optimal solutions $\boldsymbol{w}^*$ that minimize $\mathbf{L}(\boldsymbol{w})$[1]:

$$\boldsymbol{w}^* \in \operatorname*{argmin}_{\boldsymbol{w}} \mathbf{L}(\boldsymbol{w}) \tag{1}$$

**Definition 2.1.** (Pareto Optimality): A solution $\boldsymbol{w}_2$ dominates $\boldsymbol{w}_1$ iff $\mathcal{L}^m(\boldsymbol{w}_2) \leq \mathcal{L}^m(\boldsymbol{w}_1), \forall m \in \{1, \ldots, M\}$, and $\mathbf{L}(\boldsymbol{w}_1) \neq \mathbf{L}(\boldsymbol{w}_2)$. In other words, a dominating solution has a lower loss value on at least one task and no higher loss value on any task. A solution $\boldsymbol{w}^*$ is called *Pareto optimal* iff there exists no other solution dominating $\boldsymbol{w}^*$.

**Definition 2.2.** (Pareto front): The sets of Pareto optimal points and their function values are called *Pareto set* ($\mathcal{P}_{\boldsymbol{w}}$) and *Pareto front* ($\mathcal{P}_{\mathbf{L}} = \{\mathbf{L}(\boldsymbol{w})_{\boldsymbol{w} \in \mathcal{P}_{\boldsymbol{w}}}\}$), respectively.

**Linear Scalarization.** In MOO, a standard technique to solve the $M$-dimensional problem is using a preference vector $\boldsymbol{r} \in \mathcal{S} \triangleq \{\mathbb{R}^M | \sum_{m=1}^M r_m = 1, r_m \geq 0, \forall m \in \{1, \ldots, M\}\}$ in the $M$-dimensional probability simplex (Lin et al., 2019; Mahapatra & Rajan, 2020; Ruchte & Grabocka, 2021). Every $\boldsymbol{r} \in \mathcal{S}$ yields a convex combination of the loss functions in Equation 1 as

---
[1]$\boldsymbol{w}$ can be replaced with any other parameter here, also architectural ones (see Section 3).

$\mathcal{L}_r(\boldsymbol{w}) = \boldsymbol{r}^{\mathbf{T}}\mathbf{L}(\boldsymbol{w})$. Given a preference vector $\boldsymbol{r}$, one can apply standard, single-objective optimization algorithms to find a minimizer $\boldsymbol{w}_r^* = \operatorname{argmin}_{\boldsymbol{w}} \mathcal{L}_r(\boldsymbol{w})$. By sampling multiple $\boldsymbol{r}$ vectors, one can compute Pareto-optimal solutions $\boldsymbol{w}_r^*$ that profile the Pareto front. Several methods (Lin et al., 2020; Navon et al., 2021; Hoang et al., 2023; Phan et al., 2022) employ a hypernetwork (Ha et al., 2017) to generate Pareto-optimal solutions given different preference vectors as input. Similarly, in this work, we utilize a hypernetwork conditioned on scalarizations, to generate architectures.

**Multiple Gradient Descent (MGD).** MOO can be solved to local optimality via MGD (Désidéri, 2012), as a natural extension of single-objective gradient descent, which iteratively updates $\boldsymbol{w}$ towards a direction that ensures that all tasks improve simultaneously (called *Pareto improvement*): $\boldsymbol{w}' \leftarrow \boldsymbol{w} - \xi g_{\boldsymbol{w}}^*$, where $g_{\boldsymbol{w}}^*$ is a vector field that needs to be determined. If we denote by $g_{\boldsymbol{w}}^m = \nabla_{\boldsymbol{w}}\mathcal{L}^m(\boldsymbol{w})$ the gradient of the $m$-th scalar loss function, via Taylor approximation, the decreasing direction of $\mathcal{L}^m$ when we update $\boldsymbol{w}$ towards $g_{\boldsymbol{w}}^*$ is given by $\langle g_{\boldsymbol{w}}^m, g_{\boldsymbol{w}}^* \rangle \approx -(\mathcal{L}^m(\boldsymbol{w}') - \mathcal{L}^m(\boldsymbol{w}))/\xi$. In MGD $g_{\boldsymbol{w}}^*$ is chosen to maximize the slowest update rate among all objectives:

$$g_{\boldsymbol{w}}^* \propto \operatorname*{argmax}_{g_{\boldsymbol{w}} \in \mathbb{R}^d, ||g_{\boldsymbol{w}}|| \leq 1} \left\{ \min_{m \in [M]} \langle g_{\boldsymbol{w}}, g_{\boldsymbol{w}}^m \rangle \right\}. \qquad (2)$$

The early work of Désidéri (2012) has been extended in various settings, particularly multi-task learning, with great promise (Sener & Koltun, 2018; Lin et al., 2019; Mahapatra & Rajan, 2020; Liu & Vicente, 2021), but these approaches are applied to mainly a fixed architecture and extending them to a search space of architectures is non-trivial.

**One-shot NAS and Bi-Level optimization.** With the architecture space being intrinsically discrete, large and hence expensive to search on, most existing differentiable NAS approaches leverage the weight sharing paradigm and continuous relaxation to enable gradient descent (Liu et al., 2019; Pham et al., 2018; Bender et al., 2018; Xie et al., 2019; Xu et al., 2020a; Dong & Yang, 2019; Chen et al., 2021b; Liu et al., 2023; Movahedi et al., 2022; Zhang et al., 2021). Typically, in these approaches, architectures are stacks of cells, where the cell structure is represented as a directed acyclic graph (DAG) with $N$ nodes and $E$ edges. Every transition from node $i$ to $j$, i.e. edge $(i, j)$, is associated with an operation $o^{(i,j)} \in \mathcal{O}$, where $\mathcal{O}$ is a predefined candidate operation set. Liu et al. (2019) proposed a continuous relaxation of the search space by parameterizing the discrete operation choices in the DAG edges via a learnable vector $\alpha$. This allows to frame the NAS problem as a bi-level optimization one, with differentiable objectives w.r.t. all variables:

$$\operatorname*{argmin}_{\alpha} \mathcal{L}^{val}(\boldsymbol{w}^*(\alpha), \alpha)$$
$$s.t. \quad \boldsymbol{w}^*(\alpha) = \operatorname*{argmin}_{\boldsymbol{w}} \mathcal{L}^{train}(\boldsymbol{w}, \alpha), \qquad (3)$$

where $\mathcal{L}^{train}$ and $\mathcal{L}^{val}$ are the empirical losses on the training and validation data, respectively, $\boldsymbol{w}$ are the supernetwork parameters, $\alpha \in \mathcal{A}$ are the continuous architectural parameters, and $\boldsymbol{w}^*(\alpha) : \mathcal{A} \to \mathbb{R}^d$ is a best response function that maps architectures to their optimal weights.

**Comparison to single-objective constrained NAS.** Early NAS methods predominantly targeted high accuracy, whereas contemporary hardware-aware differentiable NAS approaches (Wu et al., 2019; Wan et al., 2020; Cai et al., 2018; Wu et al., 2021; Fu et al., 2020; Xu et al., 2020b; Jiang et al., 2021; Wang et al., 2021) are designed to identify architectures optimized for target hardware efficiency. Typically, these methods integrate hardware constraints within their objectives, yielding a single optimal solution and necessitating multiple search iterations to construct the Pareto front. Our proposed algorithm addresses this by profiling the entire Pareto front in a *single search iteration*. While single-objective constrained optimization is advantageous in scenarios demanding optimization of one objective under a specific constraint, practical applications often require a suite of models adaptable to varying user preferences even on a single device. For example, meeting stringent hardware or task requirements (e.g., memory usage, real-time inference) while elucidating the trade-offs (i.e., Pareto front) between precision and performance is critical, as substantial runtime reductions with minimal performance degradation may be acceptable or mitigated by other components.

## 3. Hardware-aware Multi-objective Differentiable Neural Architecture Search

We first formalize the multi-objective bi-level optimization NAS problem across multiple hardware devices, and then introduce a scalable and differentiable method that combines MGD with linear scalarizations to efficiently solve this problem.

### 3.1. Problem Definition & Sketch of Solution Approach

In multi-objective NAS, the bi-level problem described in Equation 1 becomes more difficult, since we are not only concerned with finding $\boldsymbol{w}^*$ given a fixed architecture, but we want to optimize in the space of architectures $\mathcal{A}$ as well. Assuming we have $T$ hardware devices (target functions) and $M$ objectives (e.g. accuracy, latency, etc.), similar to (3), for every $t \in \{1 \ldots T\}$, the Pareto set of the multi-objective NAS problem is obtained by solving the following bi-level optimization problem:

$$\operatorname*{argmin}_{\alpha} \mathbf{L}_t^{valid}(\boldsymbol{w}^*(\alpha), \alpha)$$
$$s.t. \quad \boldsymbol{w}^*(\alpha) = \operatorname*{argmin}_{\boldsymbol{w}} \mathbf{L}_t^{train}(\boldsymbol{w}, \alpha), \qquad (4)$$

where the $M$-dimensional loss vector $\mathbf{L}_t(\boldsymbol{w}^*(\alpha), \alpha) \triangleq \left(\mathcal{L}_t^1(\boldsymbol{w}^*(\alpha), \alpha), \ldots, \mathcal{L}_t^M(\boldsymbol{w}^*(\alpha), \alpha)\right)$ is evaluated $\forall t \in$

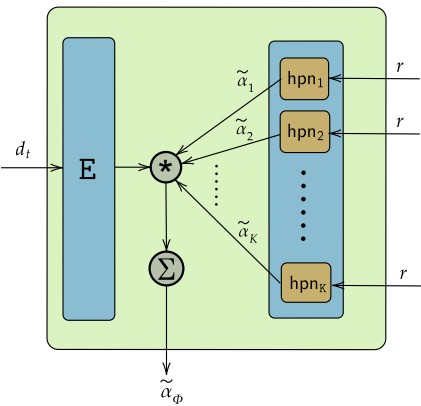

*Figure 2.* Architecture overview of the `MetaHypernetwork`, which gets as input a device embedding $d_t$ (input to an embedding layer E) and a scalarization $\boldsymbol{r}$ (input to K hypernetworks) and yields an architecture encoding $\tilde{\alpha}$.

$\{1, \dots, T\}$. $\mathbf{L}_t^{train}$ and $\mathbf{L}_t^{valid}$ are the vectors with all $M$ loss functions evaluated on the train and validation splits of $\mathcal{D}$, used in the lower- and upper-level problems of (4), respectively.

Our goal is to find Pareto-optimal architectures for each target device, covering diverse and representative preferences for different objectives. However, naively solving (4) for each device $t$ requires $T$ independent bi-level searches. To address this, we integrate a single `MetaHypernetwork` within the one-shot model (supernetwork) used in conventional NAS (Bender et al., 2018; Pham et al., 2018; Liu et al., 2019), generating architectures based on device embeddings and preference vectors in a *single search run*.

### 3.2. Algorithm Design and Components

Our search model is composed of four modular components shown in Figure 1: (1) a `MetaHypernetwork` that generates the architectural distribution; (2) an `Architect` that samples discrete architectures from this distribution; (3) a `Supernetwork` that exploits the weight sharing paradigm for search efficiency and provides a proxy for the network accuracy; and (4) a `MetaPredictor` that predicts hardware metrics and enables gradient propagation. We now discuss each of these in turn.

**`MetaHypernetwork`.** In order to generate architectures across multiple devices, inspired by Wang et al. (2022) and Lin et al. (2020), we propose a `MetaHypernetwork` that can meta-learn across different hardware devices (see Figure 2). Hypernetworks are a class of neural networks that generate the parameters of another model. They were initially proposed for model compression (Ha et al., 2017) and were later adopted for NAS (Brock et al., 2018) and MOO (Navon et al., 2021; Lin et al., 2020). Here, given a preference vector $\boldsymbol{r} = (r_1, \dots, r_M)$ and a hardware de-

vice feature vector $d_t$, for device $t \in \{1, \dots, T\}$, we use the `MetaHypernetwork` $H_\Phi(\boldsymbol{r}, d_t)$, parameterized by $\Phi$, to generate an un-normalized architecture distribution $\tilde{\alpha}_\Phi$ that is later used to compute the upper-level updates in (4). Similar to Lee et al. (2021b), $d_t$ is a fixed-size feature vector that is obtained by evaluating a fixed set of reference architectures on device $t$. The `MetaHypernetwork` is composed of 2 main components (see Figure 2):

1. *A bank* `hpn`$_1$,…,`hpn`$_K$ *of $K$ independent hypernetworks*, that parse the preference vector $\boldsymbol{r}$ and generate the architectural parameters $\tilde{\alpha}_1, \dots, \tilde{\alpha}_K$, respectively.

2. *An embedding layer* E, that learns a similarity map from device feature vectors to the bank of `hpn`s. E takes as input the device feature vector $d_t$ and outputs an attention vector of size $K$.

The final output, $\tilde{\alpha}_\Phi$, of the `MetaHypernetwork` is computed as a weighted sum of the outputs of the $K$ hypernetworks, where the vector of weights is the output of the embedding layer E. For a more detailed description of the `MetaHypernetwork` we refer the reader to Appendix F.2.

In all experiments, we initialize the `MetaHypernetwork` to yield a uniform probability mass over all architectural parameters for all scalarizations and device embeddings. By using the preference vector $\boldsymbol{r}$ to create a linear scalarization of $\mathbf{L}_t$ and the `MetaHypernetwork` to model the architectural distribution across $T$ devices, the bi-level problem in (4) reduces to:

$$\operatorname*{argmin}_{\Phi} \mathbb{E}_{\boldsymbol{r} \sim \mathcal{S}}\big[\boldsymbol{r}^{\mathbf{T}}\mathbf{L}_t^{valid}(\boldsymbol{w}^*(\alpha_\Phi), \alpha_\Phi)\big]$$
$$s.t. \quad \boldsymbol{w}^*(\alpha_\Phi) = \operatorname*{argmin}_{\boldsymbol{w}} \mathbb{E}_{\boldsymbol{r} \sim \mathcal{S}}\big[\boldsymbol{r}^{\mathbf{T}}\mathbf{L}_t^{train}(\boldsymbol{w}, \alpha_\Phi)\big], \tag{5}$$

where $\alpha_\Phi$ are the normalized architectural parameters obtained from the `Architect` $\Lambda(\tilde{\alpha}_\Phi)$ and $\boldsymbol{r}^{\mathbf{T}}\mathbf{L}_t(\cdot, \alpha_\Phi) = \sum_{m=1}^{M} r_m \mathcal{L}_t^m(\cdot, \alpha_\Phi)$ is the scalarized loss for device $t$. Conditioning the `MetaHypernetwork` on the hardware embeddings allows us to generate architectures on new test devices without extra finetuning or meta-learning steps. We use the *Dirichlet* distribution $Dir(\beta)$, $\beta = (\beta_1, \dots, \beta_M)$, to sample the preference vectors and approximate the expectation over the scalarizations using Monte Carlo sampling. In our experiments we set $\beta_1 = \dots = \beta_M = 1$, for a uniform sampling over the $(M-1)$-simplex, however, one can set these differently based on user priors or make it a learnable parameter (Chen et al., 2021b).

**`MetaPredictor`.** For the cheap-to-evaluate hardware objectives, such as latency, energy consumption, we employ a regression model $p_\theta^m(\alpha, d_t^m)$ that predicts the target labels $y_t^m$ for objective $m$ and device $t$, given an architecture $\alpha$ and device embedding $d_t^m$. We use the same

---

**Algorithm 1:** MODNAS

**Data:** $\mathcal{D}_{train}$; $\mathcal{D}_{valid}$; Supernetwork; device features $\{d_t\}_{t=1}^T$; MetaHypernetwork $H_\Phi$; nr. of objectives $M$; Architect $\Lambda$; learning rates $\xi_1, \xi_2$.

1 **while** *not converged* **do**
2    **for** $t \in \{1, \ldots, T\}$ **do**
3      Sample scalarization $\boldsymbol{r} \sim Dir(\beta)$
4      Set arch params $\tilde{\alpha}_\Phi \leftarrow H_\Phi(\boldsymbol{r}, d_t)$
5      Sample $\alpha_\Phi \sim \Lambda(\tilde{\alpha}_\Phi)$ from Architect
6      $g_\Phi^t \leftarrow \sum_{m=1}^M r_m \nabla_\Phi \mathcal{L}_t^m(\mathcal{D}_{valid}; \boldsymbol{w}, \alpha_\Phi)$
7    $\gamma \leftarrow$ FrankWolfeSolver$(g_\Phi^1, \ldots, g_\Phi^T)$ ;    // see Alg.4
8    $g_\Phi^* \leftarrow \sum_{t=1}^T \gamma_t \cdot g_\Phi^t$
9    $\Phi \leftarrow \Phi - \xi_1 \cdot g_\Phi^*$ ;        // update MetaHypernetwork
10    **for** $t \in \{1, \ldots, T\}$ **do**
11      Sample scalarization $\boldsymbol{r} \sim Dir(\beta)$
12      Set arch params $\tilde{\alpha}_\Phi \leftarrow H_\Phi(\boldsymbol{r}, d_t)$
13      Sample $\alpha_\Phi \sim \Lambda(\tilde{\alpha}_\Phi)$ from Architect
14      $g_{\boldsymbol{w}}^t \leftarrow \sum_{m=1}^M r_m \nabla_{\boldsymbol{w}} \mathcal{L}_t^m(\mathcal{D}_{train}; \boldsymbol{w}, \alpha_\Phi)$
15    $g_{\boldsymbol{w}}^* \leftarrow \frac{1}{T} \sum_{t=1}^T g_{\boldsymbol{w}}^t$
16    $\boldsymbol{w} \leftarrow \boldsymbol{w} - \xi_2 \cdot g_{\boldsymbol{w}}^*$ ;       // update Supernetwork
17 **return** $H_\Phi$

---

predictors as Lee et al. (2021b) and optimize the MSE loss: $\min_\theta \mathbb{E}_{\alpha \sim \mathcal{A}, t \sim [T]} \big(y_t^m - p_\theta^m(\alpha, d_t^m)\big)^2$, as done in Lee et al. (2021a) for meta-learning performance metrics across datasets. In our experiments, we pretrain a separate MetaPredictor for every hardware objective $m$ (e.g. latency, energy, etc.) on a subset of $(\alpha, y_t^m)$ pairs, and use its predicted value directly in (5) as $\mathcal{L}_t^m(\cdot, \alpha_\Phi) = p_\theta^m(\alpha_\Phi, d_t^m)$. During search, we freeze and do not update further the MetaPredictor parameters $\theta$.

**Supernetwork**. For expensive objectives like neural network classification accuracy, we use a Supernetwork that encodes the architecture space and shares parameters between architectures, providing a best response function $\boldsymbol{w}^*(\alpha_\Phi)$ for the scalarized loss in (5). While any parametric model could estimate this function, such as performance predictors (Lee et al., 2021a), this requires an expensive prior step of creating the training dataset for the predictor. To reduce memory costs of Supernetwork training, we: (1) use a one-hot encoding of $\alpha_\Phi$ for differentiable architecture sampling (Dong & Yang, 2019; Cai et al., 2018; Xie et al., 2019), activating only one architecture per step, and (2) entangle operation choice parameters in the Supernetwork, further increasing memory efficiency beyond weight sharing (Sukthanker et al., 2023).

**Architect.** The Architect $\Lambda(\tilde{\alpha})$ samples discrete architectural configurations from the un-normalized distribution $\tilde{\alpha}_\Phi = H_\Phi(\boldsymbol{r}, d_t)$ and enables gradient estimation through discrete variables for $\nabla_\Phi \mathbf{L}_t(\cdot, \alpha_\Phi)$. Methods such as GDAS (Dong & Yang, 2019) utilize the Straight-Through Gumbel-Softmax (STGS) estimator (Jang et al., 2017), that integrates the Gumbel reparameterization trick to approximate the gradient. Here we employ the recently proposed *ReinMax* estimator (Liu et al., 2023), that yields second-order accuracy without the need to compute second-order derivatives. See Appendix C.1 for more details on these discrete samplers. Similar to the findings in Liu et al. (2023), in our initial experiments, ReinMax outperformed the GDAS STGS estimator (see Figure 11 in the Appendix), therefore, we use ReinMax in all following experiments.

### 3.3. Optimizing the MetaHypernetwork via MGD

We denote the gradient of the scalarized loss in (5) with respect to the MetaHypernetwork parameters $\Phi$, shared across all devices $t \in 1, \ldots, T$, as: $g_\Phi^t = \boldsymbol{r}^\mathbf{T} \nabla_\Phi \mathbf{L}_t(\cdot, \alpha_\Phi) = \sum_{m=1}^M r_m \nabla_\Phi \mathcal{L}_t^m(\cdot, \alpha_\Phi)$, where $\alpha_\Phi$ is the discrete architectural sample from the Architect $\Lambda(\tilde{\alpha}\Phi)$. Multiple Gradient Descent (MGD) (Désidéri, 2012; Sener & Koltun, 2018) provides a plausible approach to estimate the update directions for every task simultaneously by maximizing (2). Via the Lagrangian duality, the optimal solution to equation 2 is $g_\Phi^* \propto \sum_{t=1}^T \gamma_t^* g_\Phi^t$, where $\{\gamma_t^*\}_{t=1}^T$ is the solution of the following minimization problem:

$$\min_{\gamma_1, \ldots, \gamma_T} \left\{ \left\| \sum_{t=1}^T \gamma_t g_\Phi^t \right\|_2^2 \Bigg| \sum_{t=1}^T \gamma_t = 1, \gamma_t \geq 0, \forall t \right\}$$

The solution to this problem is either 0 or, given a small step size $\xi$, a descent direction that monotonically decreases all objectives at the same time and terminates when it finds a Pareto stationary point, i.e. $g_\Phi^t = 0, \forall t \in \{1, \ldots, T\}$. When $T = 2$, the problem above simplifies to $\min_{\gamma \in [0,1]} \left\| \gamma g_\Phi^1 + (1 - \gamma) g_\Phi^2 \right\|_2^2$, which is a quadratic function of $\gamma$ with a closed form solution:

$$\gamma^* = \max \left( \min \Big( \frac{(g_\Phi^2 - g_\Phi^1)^\mathbf{T} g_\Phi^2}{\|g_\Phi^1 - g_\Phi^2\|_2^2}, 1 \Big), 0 \right).$$

When $T > 2$, we utilize the *Frank-Wolfe* solver (Jaggi, 2013) as in Sener & Koltun (2018), where the analytical solution in for $T = 2$ is used inside the line search. We provide the full algorithm for computing $\gamma^*$ in Algorithm 4 in Appendix C.2.

In Algorithm 1 and Figure 1 we provide the pseudocode and an illustration of the overall search phase of MODNAS. For every mini-batch sample from $\mathcal{D}_{valid}$, we iterate over the device features $d_t$ (line 2), sample one scalarization $\boldsymbol{r}$ and condition the MetaHypernetwork on both $\boldsymbol{r}$ and $d_t$ to

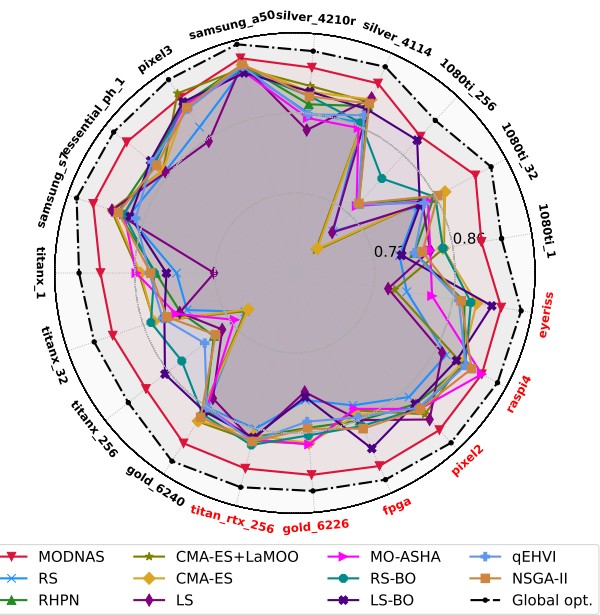

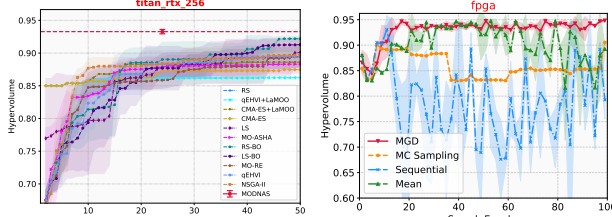

*Figure 4.* (*left*) HV over number of evaluated architectures on NAS-Bench-201 of MODNAS and the blackbox MOO baselines on a test device. Note that for MODNAS we only do 24 full evaluations. (*right*) HV over search epochs of different gradient schemes.

devices. Refer to Appendices G and H for details.

**Evaluation.** At test time, in order to profile the Pareto front with MODNAS on unseen devices, we sample 24 equidistant preference vectors $r$ from the $M$-dimensional probability simplex and pass them through the pretrained MetaHypernetwork $H_\Phi(r, d_t)$ to get 24 architectures. Here the test device feature $d_t$ is obtained similarly as for the train devices.

**Baselines.** We compare MODNAS against several baselines[2], such as Random Search (RS), Local Search (LS) and various Evolutionary Strategy and Bayesian Optimization MOO methods. Please refer to Appendix D for a more comprehensive description of each of them. Furthermore, we also evaluate the MetaHypernetwork with randomly initialized weights (RHPN).

**Metrics.** To assess the quality of the Pareto set solutions, we use the *hypervolume (HV)* indicator, which is a standard metric in MOO. Given a *reference point* $\rho = [\rho^1, \ldots, \rho^m] \in \mathbb{R}_+^M$ that is an upper bound for all objectives $\{f^m(\cdot; \boldsymbol{w}, \alpha)\}_{m=1}^M$, i.e. $\sup_\alpha f^m(\cdot; \boldsymbol{w}, \alpha) \leq \rho^m$, $\forall m \in [M]$, and a Pareto set $\mathcal{P}_\alpha \subset \mathcal{A}$, $\text{HV}(\mathcal{P}_\alpha)$ measures the region of non-dominated points bounded above from $\rho$:

$$\lambda\Big(\big\{q \in \mathbb{R}_+^M \mid \exists \alpha \in \mathcal{P}_\alpha : q \in \prod_{m=1}^M [f^m(\cdot; \boldsymbol{w}, \alpha), \rho^m]\big\}\Big),$$

where $\lambda(\cdot)$ is the Euclidean volume. HV can be interpreted as the total volume of the union of the boxes created by the Pareto front.

### 4.1. Simultaneous Pareto Set Learning across 19 devices

We firstly validate the scalability and learning capability of MODNAS by evaluating on the NAS-Bench-201 cell-based convolutional space. Here we want to optimize both latency and classification accuracy on all devices. We utilize the same set of 19 heterogeneous devices as Lee et al. (2021b), from which we use 13 for search and 6 at test time. For the latency predictor, we use the one from HELP, namely a

*Figure 3.* Hypervolume (HV) of MODNAS and baselines across 19 devices on NAS-Bench-201. For every device, we optimize for 2 objectives, namely *latency (ms)* and *test accuracy* on CIFAR-10. For each method, metric and device we report the mean of 3 independent search runs. Higher area in the radar plot indicates better HV. Test devices are colored in red around the plot.

generate the un-normalized architectural distribution $\tilde{\alpha}_\Phi$ (lines 3-4). We then compute the device-specific gradient in line 6 which is used to estimate the $\gamma$ coefficients (line 7) used from MGD to update $\Phi$ (lines 8-9). Similarly to Liu et al. (2019), we use the first-order approximation to obtain the best response function in the lower level (lines 10-14) and repeat the same procedure for the upper-level (lines 2-6), except now the Supernetwork weights $\boldsymbol{w}$ are updated with the mean gradient (line 15), over devices.

## 4. Experiments

In this section, we firstly demonstrate the scalability and generalizability of our MODNAS approach on a NAS tabular benchmark (Section 4.1). Then, we validate MODNAS on larger search spaces for Machine Translation (Section 4.2), Image Classification and Language Modeling (Section 4.3).

**Search Spaces and Datasets.** We evaluate MODNAS on 4 search spaces: (1) **NAS-Bench-201** (Dong & Yang, 2020; Li et al., 2021) with 19 devices and CIFAR-10 dataset; (2) **MobileNetV3** from Once-for-All (OFA) (Cai et al., 2020) with 12 devices and ImageNet-1k dataset; (3) **Hardware-Aware-Transformer (HAT)** (Wang et al., 2020) on the machine translation benchmark WMT'14 En-De across 3 different hardware devices; (4) **HW-GPT-Bench** (Sukthanker et al., 2024) – a GPT-2 based space used for language modeling on the OpenWebText (Gokaslan & Cohen, 2019) across 8

---

[2]We use the implementations from SyneTune (Salinas et al., 2022): https://github.com/awslabs/syne-tune

graph convolutional network (GCN), which we pretrain for 3 GPU hours on the ground truth latencies on the 13 search devices as described in Section 3. We run the MODNAS search (see Appendix F for more details on the search hyperparameters), as described in Algorithm 1, for 100 epochs (22 GPU hours on a single NVidia RTX2080Ti) and show the HV in Figure 3 of the evaluated Pareto front in comparison to the baselines, for which we allocated the same search time budget across all devices equivalent to the MODNAS search + evaluation.

Most notably, MODNAS consistently outperforms all other baselines across every device. For the baselines, we conducted 19 separate search runs (one for each device), whereas MODNAS leverages meta-learning to generate the Pareto set on each device using the same `MetaHypernetwork` in a single search run. Interestingly, the trained MODNAS attention-based `MetaHypernetwork` significantly outperforms the RHPN baseline in profiling the Pareto front, demonstrating its *effectiveness in optimizing across multiple devices and conflicting objectives simultaneously*. In Figure 14a in the Appendix, we compare MODNAS with additional baselines, running them at double the budget used for the experiments in Figure 3. Figure 4 (left) (see Figure 17 in the appendix for all devices) shows that most baselines require more than twice the number of architecture evaluations to reach the same HV as MODNAS. Results show that MODNAS remains the top performer across hardware devices on average. Furthermore, Figure 14 presents radar plots for four additional metrics.

**`MetaHypernetwork` update schemes: robustness of MGD.** We now compare the MGD update scheme for the `MetaHypernetwork` $\Phi$ (line 9 in Alg. 1) against (1) the **mean** gradient over tasks: $\Phi \leftarrow \Phi - \xi \frac{1}{T} \sum_{t=1}^{T} g_{\Phi}^t$; (2) **sequential** updates with all single tasks' gradients: $\Phi \leftarrow \Phi - \xi g_{\Phi}^t, \forall t$; (3) single updates using gradients of **MC samples** over tasks: $\Phi \leftarrow \Phi - \xi g_{\Phi}^t, t \sim \{1, \dots T\}$. Figure 4 (right) (see Figure 18 in Appendix J for more results) shows the HV over search epochs for these schemes. MGD, by accounting for inter-task dependencies, achieves higher final HV, better anytime performance, and faster convergence than the other schemes.

**Scalability to three objectives.** We now demonstrate the scalability of MODNAS to 3 objectives, namely, accuracy, latency and energy consumption. For this experiment we use the FPGA and Eyeriss tabular energy usage values from HW-NAS-Bench (Li et al., 2021). In addition to the `MetaPredictor` for latency, we pretrain a second predictor on the energy usage objective. We then run MODNAS and the MOO baselines with the same exact settings as for 2 objectives. Results shown in Figure 6 indicate that MODNAS can scale to $M > 2$ without additional search costs or hyperparameter tuning and yet achieves HV close to the

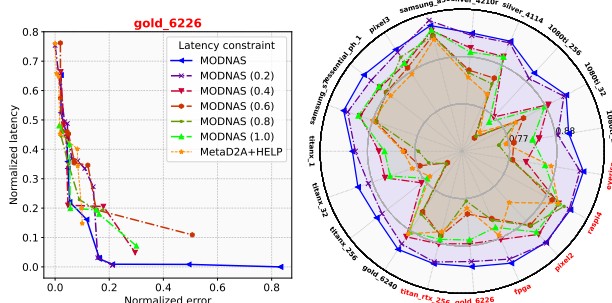

*Figure 5.* Pareto front on Eyeriss (*left*) and HV across devices (*right*) of MODNAS ran with various latency constraints on NAS-Bench 201. See Fig. 15 in Appendix I for all results.

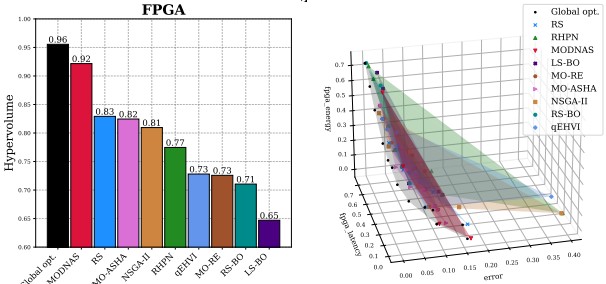

*Figure 6.* HV (*left*) and Pareto front (*right*) of MODNAS and baselines on FPGA with 3 normalized objectives: error, latency and energy usage. HV was computed using the $(1, 1, 1)$ reference point on the right 3D plot. See Fig. 12 for results on Eyeriss.

global optimum front of the NAS-Bench-201 space.

**MODNAS vs. constrained single-objective optimization.** To compare against single-objective NAS with hardware constraints in the objective, we run **MetaD2A+HELP** (Lee et al., 2021b). Since MetaD2A + HELP is not able to profile the Pareto front directly, we run the NAS search 24 times with different constraints, which we compute by denormalizing the same 24 equidistant preference vectors we use to evaluate MODNAS. We also extend MODNAS to incorporate user prior constraints over the multiple objectives being optimized during search Namely, we add a normalized constraint $c^m$, such that if the predicted value from the `MetaPredictor` during search satisfies this constraint, i.e. $p_\theta^m(\alpha_\Phi, d_t^m) \leq c^m$, we remove the gradient w.r.t. to that objective in lines 6 and 14 of Algorithm 1. In Figure 5 (other devices in Figure 15) we can see that when increasing the latency constraint to 1 (only cross-entropy optimized), though the HV decreases, MODNAS returns Pareto sets with more performant architectures. MetaD2A+HELP, though conducting multiple search runs per constraint, focuses more on highly performant architectures, and is not able to return a diverse solution set.

### 4.2. Pareto Front Profiling on Transformer Space

To demonstrate its effectiveness beyond image classification and CNN spaces, we apply MODNAS to the hardware-

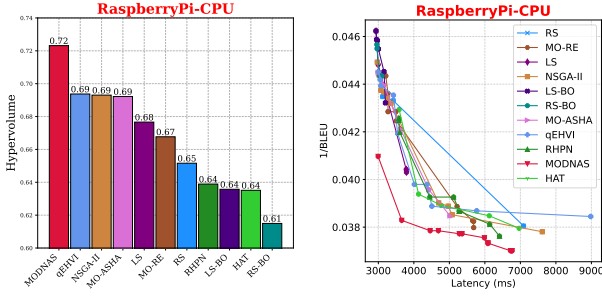

Figure 7. HV and Pareto fronts of MODNAS and baselines across devices on the HAT space.

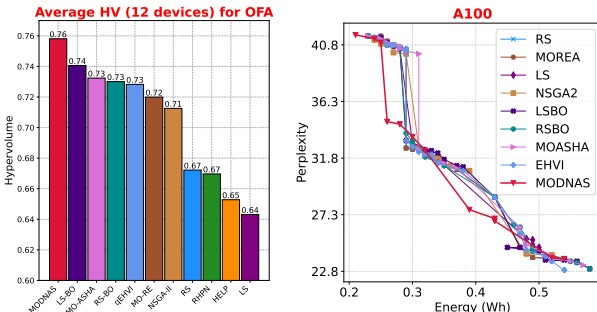

Figure 8. (*left*) Average HV of MODNAS and baselines across 12 devices on OFA space. For every device we optimize for 2 objectives, namely *latency (ms)* and *test accuracy* on ImageNet-1k. (*right*) Pareto front of MODNAS and baselines on the HW-GPT-Bench, A100 GPU.

aware Transformer (HAT) search space from Wang et al. (2020) on the WMT'14 En-De (Jean et al., 2015; Macháček & Bojar, 2014) machine translation task. We pretrained the MetaPredictor (details in Appendix F.1) for 5 GPU hours on 2000 architecture samples from the search space and then conducted the search for 110 epochs (6 days on 8 NVIDIA RTX A6000 GPUs) using 2 search devices, adhering to the same hyperparameters as Wang et al. (2020) to optimize for *latency* and *validation cross entropy loss*. Each baseline was allocated 2.5× more runtime budget than MODNAS, resulting in 1300 (RS-BO) to 6000 (MO-ASHA) total architecture evaluations, whereas MODNAS evaluated only 24 generated architectures. Details on the HAT search space and search hyperparameters are in Appendix G. We evaluated MODNAS on all 3 devices (2 search and 1 test) using the BLEU score, and results in Figure 7 show that MODNAS outperforms all baselines, achieving a higher hypervolume (left plot) of the generated Pareto fronts (right plot). For HAT, we evaluated the architectures provided in their paper. Additional results on other training devices and evaluation metrics are presented in Figures 20, 21 and 22 in the Appendix.

### 4.3. Efficient Differentiable MOO starting from Pretrained Supernetworks

**Image Classification on ImageNet-1k.** We now validate MODNAS on ImageNet-1k using the MovileNetV3 search space from Once-for-All (OFA) (Cai et al., 2020). For this experiment, we run MODNAS using 11 search (and 1 test) devices starting with the pretrained OFA supernetwork and run the search further for 1 day on 8 RTX2080Ti GPUs. During the search, we only update the MetaHypernetwork weights and keep the pretrained Supernetwork weights frozen. Details on the search space and hyperparameters are in Appendices G and F.3. We use the simple MLP from Lee et al. (2021b) as our MetaPredictor, pretraining it for 6 hours on 5000 sampled architecture-latency pairs. To evaluate the 24 points generated by our MetaHypernetwork and baselines, we use the OFA pretrained Supernetwork. Results in Figure 8 (left) show that MODNAS achieves a

higher average HV across all devices compared to baselines, which we ran for 192 hours using the OFA pretrained accuracy predictor (see Figure 25 for all results and Figure 24 for the Pareto fronts).

**Language Modeling with GPT-2.** With the rapid growth of language model sizes, it is crucial to identify transformer variants that are efficient during inference (latency) while maintaining competitive performance. We applied MODNAS to the GPT-S space from HW-GPT-Bench (Sukthanker et al., 2024), which features a non-convex Pareto front between perplexity and hardware metric objectives. Using pretrained Supernetwork weights from HW-GPT-Bench, we conducted a single 6-hour search on 4 Nvidia A100 GPUs, optimizing for energy consumption (Wh) and perplexity across 8 different GPU devices. See Appendix F for details on the MetaHypernetwork architecture and other search hyperparameters. The Supernetwork weights were kept frozen while updating the MetaHypernetwork. Figure 8 (right) shows that, within the same time budget, MODNAS matches or surpasses other MOO baselines, demonstrating its effectiveness in optimizing beyond convex Pareto fronts.

## 5. Conclusion

In this paper, we propose a novel hardware-aware differentiable NAS algorithm for profiling the Pareto front in multi-objective problems. In contrast to constraint-based NAS methods, ours can generate Pareto optimal architectures across multiple devices with a single hypernetwork that is conditioned on preference vectors encoding the trade-off between objectives. Experiments across various hardware devices (up to 19), objectives (accuracy, latency and energy usage), search spaces (CNNs and Transformers), and applications (classification, machine translation, language modeling) demonstrate the effectiveness and efficiency of our method.

## Acknowledgments

This research was partially supported by the following sources: TAILOR, a project funded by EU Horizon 2020 research and innovation programme under GA No 952215; the Deutsche Forschungsgemeinschaft (DFG, German Research Foundation) under grant number 417962828; the European Research Council (ERC) Consolidator Grant "Deep Learning 2.0" (grant no. 101045765). Robert Bosch GmbH is acknowledged for financial support. The authors acknowledge support from ELLIS and ELIZA. Funded by the European Union. Views and opinions expressed are however those of the author(s) only and do not necessarily reflect those of the European Union or the ERC. Neither the European Union nor the ERC can be held responsible for them.

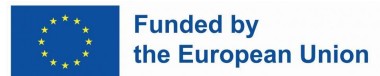

.

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

## A. Broader Impact and Implications

This work focuses on an area, neural architecture search (NAS), which has, rightfully, received criticism for its energy consumption. NAS techniques, particularly black-box techniques, require significant computational resources to train the sampled neural network architectures. Speeding up the search and training cost for different neural architectures is an important aspect of responsible research in NAS. The main goal of this work is to improve the search time in NAS, as well as the efficiency of the found architectures in terms of various hardware metrics, such as latency, energy usage, therefore reducing the energy consumption and $CO_2$ footprint.

Specifically, this paper presents a more efficient optimization procedure than previously proposed methods and achieves comparable or better solutions. Additionally, the more hardware-efficient architectures we find, correspond to more efficient neural networks, e.g. in terms of energy consumption. The energy savings of these architectures will be amplified as they might be deployed on a large number of devices. Ultimately, our method leads to decreased energy consumption during product development, as well as during the usage of the product.

## B. Extended Related Work

**Multi-objective optimization.** Multi-objective optimization (MOO) (Gunantara, 2018) is a crucial field in optimization theory, tackling decision-making scenarios with multiple conflicting objectives. MOO techniques can be categorized into gradient-based and gradient-free approaches. *Gradient-free* MOO approaches, such as evolutionary algorithms and dominance-based methods like NSGA-II (Deb et al., 2000), often suffer from sample inefficiency and are typically unsuitable for deep learning applications. On the other hand, *gradient-based* MOO methods leverage gradients. The foundational work by Désidéri (2012) has been significantly extended in multi-task learning contexts, demonstrating considerable potential (Sener & Koltun, 2018; Lin et al., 2019; Mahapatra & Rajan, 2020; Liu & Vicente, 2021). However, these methods are primarily applied to fixed architectures, and adapting them to architecture search spaces is complex. This adaptation would require retraining each architecture with multiple objectives, which is impractically expensive for large search spaces. Another major challenge in MOO is balancing the different objectives. To address this, preference vectors have been proposed to guide the prioritization of objectives on the Pareto Front (Ye & Liu, 2022; Momma et al., 2022). An emerging approach to mitigate the retraining issue involves hypernetworks, which determine the weights of the main network in MOO scenarios (Lin et al., 2020), often incorporating preference vector (Navon et al., 2021; Hoang et al., 2023; Phan et al., 2022).

**Neural Architecture Search.** A major challenge in the automated design of neural network architectures is the efficient exploration of vast search spaces. Early NAS methods relied on Reinforcement Learning (Zoph & Le, 2017), evolutionary algorithms (Deb et al., 2002; Lu et al., 2020; Elsken et al., 2019b), and other black-box optimization techniques (Daulton et al., 2022) to train and evaluate numerous architectures from scratch. The advent of one-shot NAS introduced weight sharing among architectures by training an over-parameterized network, known as a supernet, to expedite the evaluation of individual networks within the search space (Saxena & Verbeek, 2016; Bender et al., 2018; Pham et al., 2018; Liu et al., 2019). Differentiable one-shot NAS methods (Wu et al., 2019; Cai et al., 2018; Wu et al., 2021; Fu et al., 2020) further improved efficiency by applying a continuous relaxation to the search space, enabling the use of gradient descent to identify optimal sub-models within the supernet. In contrast, two-stage NAS methods initially train a supernet, often through random sampling of subnetworks, and subsequently employ black-box optimization to identify optimal subnetworks (Bender et al., 2018; Li & Talwalkar, 2020; Guo et al., 2020).

**Hardware-aware and Multi-objective Neural Architecture Search.** Early NAS methods primarily focused on maximizing accuracy for a given task. In contrast, hardware-aware NAS aims to optimize architectures for efficient performance on specific hardware devices (Benmeziane et al., 2021; Zhang et al., 2020; Lee et al., 2020; Shaw et al., 2019), naturally leading to multi-objective NAS (Hsu et al., 2018; Kim et al., 2021; Tan et al., 2019). Two-stage NAS methods can be adapted to this context by incorporating a multi-objective search in the second stage (Cai et al., 2018; Ito & Von Zuben, 2023). However, most two-stage methods depend on random sampling during supernet training, which doesn't prioritize promising architectures. Differentiable NAS methods, such as those in Wu et al. (2019); Cai et al. (2018); Wu et al. (2021); Fu et al. (2020); Xu et al. (2020b); Jiang et al. (2021); Wang et al. (2021), use latency proxies like layer-wise latencies and FLOPS (Dudziak et al., 2020) to evaluate hardware performance, combining task and hardware objectives with fixed weighting to find a single optimal solution. However, changing the objective weighting requires a complete search rerun, which is computationally demanding.

In contrast, our proposed search algorithm offers the entire Pareto Front of objectives in a single run, making it more efficient. While our focus is on multi-objective NAS for hardware constraints, our technique is applicable to other objectives such as

fairness (Martinez et al., 2020; Dooley et al., 2023; Das & Dooley, 2023), suggesting promising avenues for future research.

# C. Algorithmic components

In this section, we provide the pseudocodes for some of the algorithmic components we use in MODNAS.

## C.1. Discrete Samplers

Given the architecture parameters $\tilde{\alpha}_\Phi$ from the MetaHypernetwork, we obtain a differentiable discrete architecture sample from the Architect as $\alpha_\maltese \leftarrow \pi - \texttt{stop\_g}(\pi) + \alpha_\maltese$, where $\alpha_\maltese \sim \texttt{Cat}\big(\texttt{softmax}_1(\tilde{\alpha}_\maltese)\big)$ and

$$\pi \leftarrow 2 \cdot \texttt{softmax}_1\Big(\texttt{stop\_g}\big(\ln(\frac{\alpha_\maltese + \texttt{softmax}_\tau(\tilde{\alpha}_\maltese)}{2}) - \tilde{\alpha}_\maltese\big) + \tilde{\alpha}_\maltese\Big) - \frac{\texttt{softmax}_1(\tilde{\alpha}_\maltese)}{2}.$$

Here, Cat is the categorical distribution, $\tau$ is the temperature in the tempered softmax $\texttt{softmax}_\tau(\alpha)_i = \frac{exp(\alpha_i/\tau)}{\sum_{j=1}^{|\mathcal{O}|} exp(\alpha_j/\tau)}$, and $\texttt{stop\_g}(\cdot)$ duplicates its input and detaches it from backpropagation. Refer to the ReinMax paper (Liu et al., 2023) for more details. The algorithm pseudocode on how a one-hot encoded (discrete) architecture is sampled given an architectural unnormalized distribution $\tilde{\alpha}$ is given in Algorithm 2 and Algorithm 3, for the Straight-Through (Jang et al., 2017) and ReinMax (Liu et al., 2023) gradient estimators, respectively.

---

**Algorithm 2:** $\texttt{Straight} - \texttt{Through}$ (Jang et al., 2017)

**Data:** $\tilde{\alpha}$: softmax input, $\tau$ : temperature
**Result:** $\alpha$: one-hot samples
1   $\pi_0 \leftarrow \texttt{softmax}_1(\tilde{\alpha})$
2   $\alpha \sim \texttt{Cat}(\pi_0)$
3   $\pi_1 \leftarrow \texttt{softmax}_\tau(\tilde{\alpha})$
4   $\alpha \leftarrow \pi_1 - \texttt{stop\_g}(\pi_1) + \alpha$
5   **return** $\alpha$

---

**Algorithm 3:** $\texttt{ReinMax}$ (Liu et al., 2023)

**Data:** $\tilde{\alpha}$: softmax input, $\tau$ : temperature
**Result:** $\alpha$: one-hot samples
1   $\pi_0 \leftarrow \texttt{softmax}_1(\tilde{\alpha})$
2   $\alpha \sim \texttt{Cat}(\pi_0)$
3   $\pi_1 \leftarrow \frac{\alpha + \texttt{softmax}_\tau(\tilde{\alpha})}{2}$
4   $\pi_1 \leftarrow \texttt{softmax}_1\big(\texttt{stop\_g}\big(\ln(\pi_1) - \tilde{\alpha}\big) + \tilde{\alpha}\big)$
5   $\pi_2 = 2 \cdot \pi_1 - \frac{1}{2} \cdot \pi_0$
6   $\alpha \leftarrow \pi_2 - \texttt{stop\_g}(\pi_2) + \alpha$
7   **return** $\alpha$

---

## C.2. Frank-Wolfe Solver

In this section, we provide the pseudocode of the Frank-Wolfe solver (Jaggi, 2013) used to compute the gradient coefficients used for the MGD updates. To solve the constrained optimization problem, the Frank-Wolfe solver uses analytical solution for the line search with $T = 2$ (Algorithm 5).

---

**Algorithm 4:** $\texttt{FrankWolfeSolver}$ (Jaggi, 2013)

**Data:** $g_\Phi^1, \ldots, g_\Phi^T$
**Result:** $\gamma = (\gamma_1, \ldots, \gamma_T)$
1   Initialize $\gamma \leftarrow (\frac{1}{T}, \ldots, \frac{1}{T})$
2   Precompute $\mathcal{M}$ s.t. $\mathcal{M}_{i,j} = (g_\Phi^i)^{\mathbf{T}}(g_\Phi^j)$
3   **repeat**
4     $\hat{t} \leftarrow \text{argmin}_r \sum_{t=1}^{T} \gamma_t \mathcal{M}_{rt}$
5     $e_{\hat{t}} \leftarrow \mathcal{M}_{\hat{t},\cdot}$ ;              // $\hat{t}$-th row of $\mathcal{M}$
6     $\hat{\delta} \leftarrow \text{argmin}_\delta \big((1-\delta)\gamma + \delta e_{\hat{t}}\big)^{\mathbf{T}} \mathcal{M}\big((1-\delta)\gamma + \delta e_{\hat{t}}\big)$ ;     // using Algorithm 5
7     $\gamma \leftarrow (1-\hat{\delta})\gamma + \hat{\delta}e_{\hat{t}}$
8   **until** $\hat{\delta} \sim 0$ *or Number of Iterations Limit*;
9   **return** $\gamma$

---

**Algorithm 5:** Solver $\min_{\delta \in [0,1]} ||\delta\theta + (1-\delta)\bar{\theta}||_2^2$

**1** **if** $\theta^{\mathbf{T}}\bar{\theta} \geq \theta^{\mathbf{T}}\theta$ **then**
**2** $\quad \delta \leftarrow 1$
**3** **else if** $\theta^{\mathbf{T}}\bar{\theta} \geq \bar{\theta}^{\mathbf{T}}\bar{\theta}$ **then**
**4** $\quad \delta \leftarrow 0$
**5** **else**
**6** $\quad \delta \leftarrow \frac{(\bar{\theta}-\theta)^{\mathbf{T}}\bar{\theta}}{||\theta-\bar{\theta}||_2^2}$
**7** **return** $\delta$

## D. Multi-objective NAS algorithms

This section elaborates on the multi-objective NAS methods we utilize as baselines in Section 4.

- **Random Search (RS)** is a robust baseline for both single-objective (Bergstra & Bengio, 2012; Li & Talwalkar, 2020) and multi-objective (Cai et al., 2020; Chen et al., 2021a) architecture searches. This baseline involves randomly sampling architectures from the search space and computing the Pareto front from these samples. While RS is computationally efficient and often effective, it may not always find the optimal architectures, especially in larger search spaces.

- **Local Search (LS)** is adapted to refine solutions near Pareto-optimal points in multi-objective optimization, iteratively improving solutions within defined neighborhoods.

- **Multi-objective Asynchronous Successive Halving (MO-ASHA)** (Schmucker et al., 2021) is a multi-fidelity method that utilizes an asynchronous successive halving scheduler (Li et al., 2018) and non-dominating sorting for budget allocation. MO-ASHA uses the NSGA-II selection mechanism and the $\epsilon$-net (Salinas et al., 2021) exploration strategy that ranks candidates in the same Pareto set by iteratively selecting the one with the largest Euclidian distance from the previous set of candidates.

- **Multi-Objective Regularized Evolution (MO-RE)** builds on Regularized Evolution (RE) (Real et al., 2019), which evolves a population of candidates through mutation and periodically removes the oldest individuals, thus regularizing the population. MO-RE adapts this by using multi-objective non-dominated sorting to score candidates, with parents sampled based on these scores.

- **Non-dominated Sorting Genetic Algorithm II (NSGA-II)** (Deb et al., 2002) is a multi-objective evolutionary algorithm designed to find a Pareto set of architectures. It ranks architectures using non-dominated sorting and maintains diversity with crowding distance. Through selection, crossover, and mutation, NSGA-II evolves populations towards the Pareto front, although it is known for being sample inefficient.

- **Covariance Matrix Adaptation Evolution Strategy (CMA-ES)** (Igel et al., 2007) is an evolutionary algorithm particularly effective in continuous optimization problems. In a multi-objective context, it adapts its covariance matrix to the shape of the search space, iteratively updating its sampling distribution to favor promising regions. This method efficiently handles complex, non-linear optimization landscapes and can be adapted to multi-objective scenarios by using techniques such as Pareto-based selection to maintain a diverse set of solutions.

- **Latent Action MOO (LaMOO)** (Zhao et al., 2022) uses a parametric model and Monte Carlo Tree Search (MCTS) to learn to partition the objective space based on the dominance number, which indicates the vicinity of a point to the Pareto front relative to the other samples. qEHVI+LaMOO and CMA-ES+LaMOO use the original qEHVI and CMA-ES, respectively, as an inner routine in the learned subspaces.

- **Bayesian Optimization with Random Scalarizations (RS-BO)** (Paria et al., 2020) uses an acquisition function based on random linear scalarizations of objectives across multiple points to find the Pareto-optimal set that minimizes Bayesian regret.

- **Bayesian Optimization with Linear Scalarizations (LS-BO)** is similar to RS-BO but optimizes a single objective derived from a fixed linear combination of two objectives instead of using randomized linear scalarizations.

- **Expected Hypervolume Improvement (qEHVI)** (Daulton et al., 2020) is a Bayesian optimization acquisition function that explores the Pareto front by quantifying potential hypervolume improvement. This approach measures the volume dominated by Pareto-optimal solutions and guides the search towards regions likely to offer better trade-offs, aiding in the discovery of diverse Pareto-optimal solutions.

# E. Evaluation Details

## E.1. Other Metrics

For NAS-Bench-201, in addition, we evaluate the *generational distance* (GD) and *inverse generational distance* (IGD) (see Appendix E). See Figure 14 for the results complementary to the hypervolume radar plot in Figure 3 of the main paper.

**Generational Distance ($GD$) and Inverse Generational Distance ($IGD$).** Given a *reference set* $\mathcal{S} \subset \mathcal{A}$ and a Pareto set $\mathcal{P}_\alpha \subset \mathcal{A}$ with $dim(\mathcal{A}) = K$, the GD indicator is defined as the distance between every point $\alpha \in \mathcal{P}_\alpha$ and the closest point in $s \in \mathcal{S}$, averaged over the size of $\mathcal{P}_\alpha$:

$$GD(\mathcal{P}_\alpha, \mathcal{S}) = \frac{1}{|\mathcal{P}_\alpha|} \left( \sum_{\alpha \in \mathcal{P}_\alpha} \min_{s \in \mathcal{S}} d(\alpha, s)^2 \right)^{1/2},$$

where $d(\alpha, s) = \sqrt{\sum_{k=1}^{K} (\alpha_k - s_k)^2}$ is the Euclidean distance from $\alpha$ to its nearest reference point in $\mathcal{S}$.

The inverted generational distance (IGD) is computed as $IGD(\mathcal{P}_\alpha, \mathcal{S}) = GD(\mathcal{S}, \mathcal{P}_\alpha)$.

**Generational Distance Plus ($GD^+$) and Inverse Generational Distance Plus ($IGD^+$).** $GD^+(\mathcal{P}_\alpha, \mathcal{S}) = IGD^+(\mathcal{S}, \mathcal{P}_\alpha)$ replaces the euclidean distance $d(\alpha, s)$ in GD with:

$$d^+(\alpha, s) = \sqrt{\sum_{k=1}^{K} (\max\{\alpha_k - s_k, 0\})^2}$$

## E.2. MODNAS-SoTL

On the NAS-Bench-201 search space, since the architectures evaluated with the supernetwork weights are not highly correlated to the ones trained independently from scratch, we employ the Sum of Training Losses (SoTL) proxy from Ru et al. (2021). To profile the Pareto front with SoTL, we firstly evaluate the 24 architectures using the exponential moving average of the sum of training losses for the initial 12 epochs of training as $\sum_{e=1}^{12} 0.9^{12-e} \mathcal{L}^{train}(\boldsymbol{w}, \alpha)$, and then train from scratch only the subset of architectures in the Pareto set built using the SoTL evaluations. We present the results of MODNAS-SoTL in Figure 14, where we compare to the other baselines as well. As we see, we can further decrease the evaluation cost via MODNAS-SoTL, by trading off the number of solutions in the Pareto set with HV.

# F. Experimental Details

## F.1. `MetaPredictor` Architectures

For all search spaces we set the dimensionality of the hardware embedding to 10. This corresponds to latency evaluations on a set of 10 reference architectures, which are the same used by Lee et al. (2021b).

**NAS-Bench-201.** For the NAS-Bench-201 (Dong & Yang, 2020) search space we use a Graph Convolutional Network (GCN) as proposed in Dudziak et al. (2020). Furthermore, in addition to the *one-hot operation encoding* and *adjacency matrix* corresponding to the architecture cells, we also input the hardware embedding to this predictor, as done by Lee et al. (2021b). The number of nodes in the GCN is 8 and the dimensionality of the layers is set to 100 following HELP (Lee et al., 2021b).

**MobileNetV3 (OFA).** Following HELP (Lee et al., 2021b), we employ a simple feedforward neural network in the MobileNetV3 search space. The input dimension of the `MetaPredictor` is set to 160, matching the concatenated architecture

encoding dimension. We set the size of the hidden layers to 100. Specifically, the `MetaPredictor` comprises 2 linear layers with ReLU activation for processing the 160-dimensional one-hot architecture encoding and 2 linear layers for processing the hardware embedding. The outputs from these two paths are concatenated and passed through a final linear layer to predict the latency.

**Seq-Seq Transformer (HAT).** HELP [3] does not release the architecture or the meta-learned pretrained predictor for HAT(Wang et al., 2020). However, HAT [4] releases code and pretrained models for each of the devices and tasks trained independently. Hence, we build our single per-task `MetaPredictor` based on the architecture of the HAT predictor, i.e. a simple feedforward neural network. The input dimension corresponds to the one-hot architecture encoding of the candidate Transformer architecture. Additionally, to condition on the hardware embedding, we include 2 extra linear layers for processing the hardware embedding, which is then concatenated with the processed architecture encoding to produce the final latency prediction. The hidden dimension of the `MetaHypernetwork` is set to 400, with 6 hidden layers. The predictor's input feature dimension is 130.

**HW-GPT-Bench** We utilize the raw energy observations released in (Sukthanker et al., 2024) to train a single hardware-aware meta-predictor across energy observations from eight GPU types. Our meta-predictor is a simple MLP, similar to the one in HAT, with 4 hidden layers, 2 layers for processing the hardware embedding (which the network is conditioned on). The MLP's hidden dimension is 256, and the input feature dimension matches the one-hot encoded architecture feature map for this space, i.e., 80.

### F.2. `MetaHypernetwork` Architecture

Given a preference vector $r \in \mathbb{R}^M$, we use the hypernetwork $h_\phi(r) : \mathbb{R}^M \to \mathcal{A}$, parameterized by $\phi \in \mathbb{R}^n$, to generate an un-normalized architecture distribution $\tilde{\alpha}$ that is later used to compute the upper-level updates in (4). In our experiments, $h_\phi$ is composed of $M-1$ [5] embedding layers $e^m$, $m \in \{2, \ldots, M\}$ with $n_m$ possible learnable vectors of size $\frac{dim(\mathcal{A})}{M-1}$. The output of $h_\phi$ is the concatenation of all $M-1$ outputs of $e^m$, such that its size matches $dim(\mathcal{A})$. See Figure 2 for details.

In order to enable the hypernetwork to generate architectures across multiple devices, inspired by Wang et al. (2022) and Lin et al. (2020), we propose a `MetaHypernetwork` $H_\Phi(r, d_t) : \mathbb{R}^M \times \mathcal{H}^{M-1} \to \mathcal{A}$

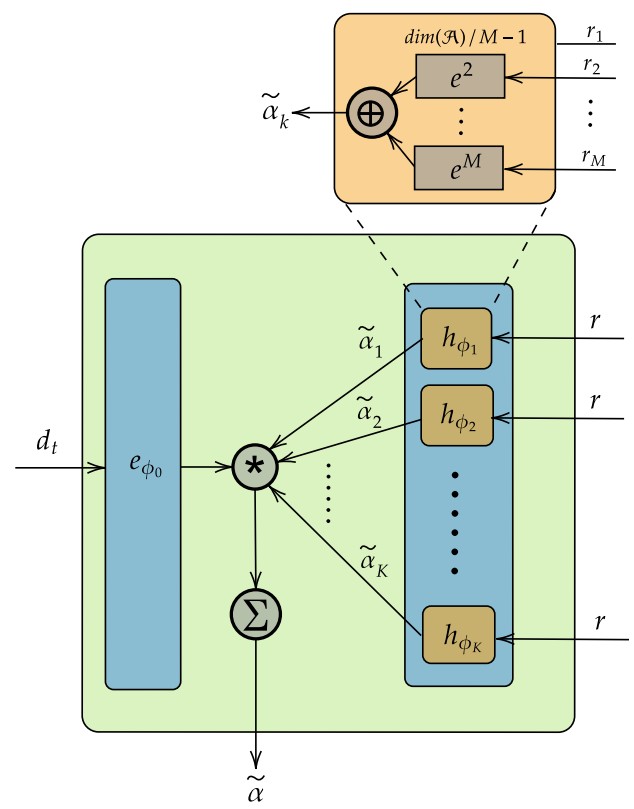

*Figure 9.* `MetaHypernetwork` architecture overview in the case of $M$ objectives. Note that $m=1$ is reserved for the accuracy objective, which we model through the cross-entropy loss in the `Supernetwork`. The initial embedding layer $e_{\phi_0}$ gets the $d_t$ hardware embedding and outputs a weight that scales each of the $K$ hypernetworks' (orange boxes) outputs from the hypernetwork bank. The scaled architectural parameters are then summed up element-wise. All individual hypernetwork $h_{\phi_k}$ get as input the same scalarization $r$. Each of them has $M-1$ embedding layers with dimensions $n_m \times \frac{dim(\mathcal{A})}{M-1}$, $\forall m \in \{2, \ldots, M\}$ that gets as input the scalarizations for objectives $m = 2, \ldots, m = M$, and yields a vector of size $\frac{dim(\mathcal{A})}{M-1}$. The output from the $M-1$ embedding layers are concatenated to give the architecture encoding $\tilde{\alpha}$.

that can meta-learn across $T$ different hardware devices (see Figure 1). The input to $H_\Phi$ is a concatenation of device feature vectors across all metrics, i.e. $d_t = \oplus_{m=2}^{M} d_t^m$. Similar to Lee et al. (2021b), $d_t^m \in \mathcal{H}$ is a fixed-size feature

---

[3] https://github.com/HayeonLee/HELP

[4] https://github.com/mit-han-lab/hardware-aware-transformers

[5] $m = 1$ (CE loss) does not have an hardware embedding.

vector representative of device $t \in \{1, \ldots, T\}$ and objective $m \in \{2, \ldots, M\}$, that is obtained by evaluating a fixed set of reference architectures for a given metric. The MetaHypernetwork, with $\Phi = \cup_{k=0}^{K} \phi_k$ parameters, contains a bank of $K > T$ hypernetworks $\{h_{\phi_k}(\boldsymbol{r})\}_{k=1}^{K}$ and an additional embedding layer $e_{\phi_0}(d_t) : \mathcal{H}^{M-1} \to \mathbb{R}^K$ at the beginning, that learns a similarity map for every device feature to the hypernetworks' bank. If we denote by $h_{\phi_{1:k}} = (h_{\phi_1} \cdots h_{\phi_k})^{\mathbf{T}}$ the vector of all hypernetworks in the bank, then, given a preference vector $\boldsymbol{r}$, to obtain $\tilde{\alpha}$ for device $t$, we compute a weighted mixture of predictions of all $h_\phi$ in the hypernetwork bank as follows:

$$\tilde{\alpha}_\Phi = H_\Phi(\boldsymbol{r}, d_t) = \sum_{k=1}^{K} e_{\phi_0}(d_t)[k] \cdot h_{\phi_k}(\boldsymbol{r}) = e_{\phi_0}(d_t) \cdot h_{\phi_{1:k}}(\boldsymbol{r}).$$

We keep the MetaHypernetwork architecture similar across search spaces. The only thing we adapt is the output dimensionality of the hypernetwork (in the hypernetwork bank of MetaHypernetwork), which corresponds to the dimensionality of the architecture parameters of the respective search space. We set the size of the initial hardware embedding layer and the hypernetwork bank to 50 for all search spaces. Furthermore, each hypernetwork has 100 possible learnable embeddings $e^m$, for every objective $m \in \{2, \ldots, M\}$, to map the scalarization vector to an architecture. See Figure 2 for an illustration of the MetaHypernetwork architecture.

For the **NAS-Bench-201** search space, we use a single embedding layer of dimensionality 30, i.e. corresponding to the dimensionality of the architecture space: $6 \times 5$ (6 edges and 5 operation choices on each edge). For the 3-objective experiment, we include an additional embedding for the energy usage objective, concatenated with the latency embedding before passing it to the MetaHypernetwork. The individual hypernetworks in the MetaHypernetwork bank have 2 embedding layers with dimensionality 15, whose outputs are concatenated to match the architecture space dimensions.

In the **MobileNetV3** space, we use 4 embedding layers – for depth, expansion ratio, kernel size, and resolution. The space comprises 5 blocks, each with 3 depth choices, making the depth embedding layer dimensionality $5 \times 3$. The kernel and expansion embedding layers have dimensions $5 \times 4 \times 3$, corresponding to 5 blocks with a maximum depth of 4 and 3 possible kernel size or expansion ratio choices. The resolution embedding layer has a dimension of 25, representing 25 possible resolution choices.

In the **Seq-Seq Transformer (HAT)** space, the individual hypernetworks of the MetaHypernetwork utilize 9 embedding layers (the encoder layer count is fixed; see Table 1):

- 2 embedding layers of size 2 for the encoder and decoder blocks to map the scalarization to the embedding dimension architecture parameter, held constant throughout the encoder or decoder block.

- 2 embedding layers with dimensions $6 \times 3$ (6 encoder/decoder layers, 3 choices) for the linear layer size in every attention block for both encoder and decoder.

- 2 embedding layers with dimensions $6 \times 2$ for the number of heads in each attention block.

- 1 embedding layer of size 6 to encode the 6 possible choices for the number of layers in the decoder.

- 1 embedding layer of size $6 \times 3$ (6 encoder layers, 3 choices) for the arbitrary encoder layer choice for attention.

- 1 embedding layer of size $6 \times 2$ (6 encoder layers, 2 choices) for the number of heads in the encoder-decoder attention.

For the **HW-GPT-Bench** space, the individual hypernetworks of the MetaHypernetwork contain 5 embedding layers:

- 1 embedding layer of dimension $1 \times 3$ for mapping the scalarization to the embedding dimension architecture parameter of the language model, with 3 choices.

- 1 embedding layer of dimension $1 \times 3$ for mapping the scalarization to the layer number dimension architecture parameter of the language model, with 3 choices.

- 1 embedding layer of dimension $12 \times 3$ for mapping the scalarization to the mlp_ratio dimension architecture parameter of the language model, with 12 layers and 3 mlp_ratio choices per layer.

- 1 embedding layer of dimension $12 \times 3$ for mapping the scalarization to the num_heads dimension architecture parameter of the language model, with 12 layers and 3 choices per layer.

- 1 embedding layer of dimension 2 for toggling the bias in linear layers on or off.

### F.3. MODNAS Hyperparameter Configurations

In Table 4, we show the search hyperparameters and their corresponding values we use to conduct our experiments with MODNAS. For the convolutional spaces we subtract a cosine similarity penalty from the scalarized loss following (Ruchte & Grabocka, 2021):

$$g_{\Phi}^t \leftarrow \boldsymbol{r}^{\mathbf{T}} \nabla_{\Phi} \mathbf{L}_t(\mathcal{D}_{valid}, \boldsymbol{w}, \alpha_{\Phi}) - \lambda \nabla_{\Phi} \frac{\boldsymbol{r}^{\mathbf{T}} \mathbf{L}_t(\mathcal{D}_{valid}, \boldsymbol{w}, \alpha_{\Phi})}{||\boldsymbol{r}|| \; ||\mathbf{L}_t(\mathcal{D}_{valid}, \boldsymbol{w}, \alpha_{\Phi})||}, \tag{6}$$

where $|| \cdot ||$ is the $l_2$ norm. We set $\lambda$ to 0.001. Empirically we did not observe significant differences on disabling the cosine penalty term.

### F.4. Normalization of objectives

Since our method relies on a *scalarization* of different objectives, it is important that the objectives being optimized are on the same scale. For simplicity, lets consider the scenario where the two objectives of interest are the *cross-entropy* loss and *latency*. Since we pretrain and freeze our MetaPredictor, the latency-scale remains constant throughout the search, while the cross-entropy loss of the Supernetwork (likely) decreases over time. To this end, we use the following max-min normalization to normalize the objectives:

$$\mathcal{L}_t^m(\cdot, \alpha_{\Phi}) = \frac{\mathcal{L}_t^m(\cdot, \alpha_{\Phi}) - \min(\bar{\mathbf{L}})}{\max(\bar{\mathbf{L}}) - \min(\bar{\mathbf{L}})}, \tag{7}$$

where $\bar{\mathbf{L}} = \bigcup_{i=1}^N \mathtt{stop\_g}\big(\mathcal{L}_t^m(\cdot, \alpha_i)^i\big)$ is the set of losses evaluated on $N$ architectures and potentially $N$ previous steps. For the latency objective, we precompute these sample-statistics using N samples (ground-truth for NAS-Bench-201 and predicted for OFA and HAT spaces) from the search space, whilst for the cross-entropy loss we compute them throughout the search. Furthermore, to take into account the decreasing cross-entropy, we reset the cross-entropy loss statistics after every epoch.

## G. Details on Search Spaces

**NAS-Bench-201** (Dong & Yang, 2020) is a convolutional, cell-based search space. The search space consists of 3 stages, each with number of channels 16, 32 and 64, respectively. Each stage contains a convolutional cell repeated 5 times. Here, every cell is represented as a directed acyclic graph (DAG) which has 4 nodes, densely connected with 6 edges. Each edge has 5 possible operation choices: a skip connection, a zero operation, a $3 \times 3$ convolution, a $5 \times 5$ convolution or an average pooling operation. NAS-Bench-201 is a tabular benchmark exhaustively constructed, where the objective is finding the optimal cell for the given macro skeleton.

*Table 1.* Encoder-Decoder Search Space for HAT.

| Module | Searchable Dim | Choices |
|---|---|---|
| **Encoder** | No. of Layers | [6] (fixed) |
| | Embedding dim | [640, 512] |
| | No. of heads | [8, 4] |
| | FFN dim | [3072, 2048, 1024] |
| **Decoder** | No. of layers | [6, 5, 4, 3, 2, 1] |
| | Embedding dim | [640, 512] |
| | No. of heads | [8, 4] |
| | FFN dim | [3072, 2048, 1024] |
| | Arbitrary-Encoder-Layer | [-1, 1, 2] |
| | Enc-Dec attention num heads | [8, 4] |

**MobileNetV3** proposed in OFA (Cai et al., 2020) is a macro convolutional search space. The different searchable dimensions in the search space are the depth (per block), the kernel size (for every layer in every block) and the channel expansion ratio (for every layer in every block). There are a total of 5 blocks, each with 3 possible depth choices and every layer in this block has 3 possible kernel sizes and channel expansion ratio choices. This amounts to a total search space size of $((3 \times 3)^2 + (3 \times 3)^3 + (3 \times 3)^4)^5 \approx 2 \times 10^{19}$. Additionally, every architecture has 25 possible choices for the size of the input resolution. The 3 possible choices for depth, kernel size and expansion ratio are $\{2, 3, 4\}$, $\{3, 5, 7\}$ and $\{3, 4, 6\}$, respectively. The input resolution choices are $\{128, 132, 136, 140, 144, 148, 152, 156, 160, 164, 168, 172, 176, 180, 184, 188, 192, 196, 200, 204, 208, 212, 216, 220, 224\}$. We use a width factor of 1.2 similar to OFA (Cai et al., 2020).

**Seq-Seq Encoder-Decoder Transformer (HAT)** (Wang et al., 2020) for the En-De machine translation task has a searchable number of layers, embedding dimension, feedforward expansion layer dim per-layer, number of heads per-layer for both

the encoder and the decoder sub-modules. In addition to this, the number of encoder layers the decoder attends to, and the number of attention heads in the encoder-decoder attention is also searchable. We present the details of the search space in Table 1.

**HW-GPT-Bench** (Sukthanker et al., 2024) is a decoder-only transformer space designed for autoregressive language modeling. The search space includes choices for embedding dimensions {768, 384, 192}, the number of layers from {10, 11, 12}, the MLP expansion ratio per layer from {2,3,4}, the number of heads per layer from {12,8,4}, and the option to toggle the bias parameter on or off in the layers.

## H. Datasets and Devices

This section describes the hardware devices and tasks used to evaluate MODNAS and the MOO baselines throughout the paper. We assess our methods across small- and large-scale image classification datasets, including CIFAR-10 and ImageNet-1K. For the machine translation task, we evaluate our method on the WMT'14 En-De dataset (Macháček & Bojar, 2014), and we use the OpenWebText (Gokaslan & Cohen, 2019) dataset for language modeling. Furthermore, we evaluate MODNAS across 19 devices on NAS-Bench-201, 12 devices on MobileNetV3, three devices on Seq-Seq Transformer, and eight devices from HW-GPT-Bench (Sukthanker et al., 2024), with zero-shot generalization to test devices. Table 2 lists the devices used. For more details on the devices, we refer readers to Lee et al. (2021b), Cai et al. (2020), Wang et al. (2020), Li et al. (2021), and Sukthanker et al. (2024).

*Table 2.* Search-test split for hardware devices and datasets for different search spaces.

| Search Space | Train-devices | Test devices | Dataset |
|---|---|---|---|
| **NAS-Bench-201** | 1080ti_1, 1080ti_32, 1080ti_256, silver_4114, silver_4210r, samsung_a50, pixel3, essential_ph_1, samsung_s7, titanx_1, titanx_32, titanx_256, gold_6240 | titan_rtx_256, gold_6226, fpga, pixel2, raspi4, eyeriss | CIFAR10 |
| **MobileNetV3 (OFA)** | 2080ti_1, 2080ti_32, 2080ti_64, titan_xp_1, titan_xp_32, titan_xp_64, v100_1, v100_32, v100_64, titan_rtx_1, titan_rtx_32 | titan_rtx_64 | ImageNet-1k |
| **Seq-Seq Transformer (HAT)** | titanxp gpu, cpu xeon | cpu raspberrypi | WMT14.en-de |
| **HW-GPT-Bench** | a40, v100, rtx2080, rtx3080 | a100, h100, P100, a6000 | OpenWebText |

## I. Runtime Comparison

In Table 3 we provide the number of GPU hours we ran MODNAS and baselines on every search space. We ran the search on NAS-Bench-201, OFA, together with the evaluations on Nvidia RTX2080Ti, while for HAT we used NVidia A6000. For both OFA and HAT, we used 8 GPUs in parallel. Similar as in Sukthanker et al. (2024), on the HW-GPT-Bench space we ran the MODNAS search and evaluations on 4 Nvidia A100 GPUs.

*Table 3.* Total amount of GPU hours required to run MODNAS' and baselines' search on every search space.

| Search Spaces | Method | Lat/En/Mem Pred. | Supernet | Acc./Ppl Pred. | Search | Total Time |
|---|---|---|---|---|---|---|
| NASBench201 | MetaD2A+HELP | 25 | - | 8629 | 0.3 | 8654.3 |
| | MOO Baselines | - | - | - | 370.5 | 370.5 |
| | MODNAS | 3 | 22 | - | 0.05 | **25.25** |
| Once-For-All | OFA+HELP | 6 | 1200 | 356 | 10 | 1572 |
| | MOO Baselines | 6 | 1200 | 356 | 192 | 1754 |
| | MODNAS | 6 | 1392 | - | 0.05 | **1398.25** |
| HAT | HAT | 15 | 346.7 | - | 210.9 | 572.6 |
| | MOO Baselines | 15 | 346.7 | - | 576 | 937.7 |
| | MODNAS | 5 | 576 | - | 0.05 | **581.25** |
| HW-GPT-Bench | MOO Baselines | 1 | 192 | - | 48 | 241 |
| | MODNAS | 1 | 216 | - | 0.05 | **217.25** |

## I.1. Computational Complexity.

Ignoring the cost to train final architectures in the Pareto set, methods like MetaD2A + HELP (Lee et al., 2021a;b) have a worst-case time complexity of $\mathcal{O}(CT)$ to build the Pareto set, where T is the number of devices and C is the number of constraints. MODNAS reduces this to $\mathcal{O}(1)$ by conditioning a single `MetaHypernetwork` on both device types and constraints. Methods like LEMONADE (Elsken et al., 2019a) and ProxylessNAS (Cai et al., 2018) apply

*Table 5.* Cost of MODNAS compared to other methods. N is the number of trained architectures during search, T is the number of devices and C is the number of constraints.

| Method | Search Cost | Pareto Set Build Cost |
|---|---|---|
| LEMONADE (Elsken et al., 2019a) | $\mathcal{O}(NT)$ | $\mathcal{O}(1)$ |
| Blackbox MOO (Deb et al., 2002; Daulton et al., 2020; Zhao et al., 2022) | $\mathcal{O}(NT)$ | $\mathcal{O}(1)$ |
| ProxylessNAS (Cai et al., 2018) | $\mathcal{O}(CT)$ | $\mathcal{O}(1)$ |
| MetaD2A + HELP (Lee et al., 2021a;b) | $\mathcal{O}(N)$ | $\mathcal{O}(CT)$ |
| OFA (Cai et al., 2020) + HELP (Lee et al., 2021b) | $\mathcal{O}(1)$ | $\mathcal{O}(CT)$ |
| **MODNAS (Ours)** | $\mathcal{O}(1)$ | $\mathcal{O}(1)$ |

constraints during the search phase, requiring an independent search per device. Black-box methods such as LEMONADE, NSGA-II (Deb et al., 2002), or qEHVI (Daulton et al., 2020) train $\mathcal{O}(NT)$ architectures or a surrogate based on $\mathcal{O}(N)$ architectures in the case of MetaD2A + HELP. In contrast, MODNAS and OFA have a cost of $\mathcal{O}(1)$ as they train a single supernetwork. Although MODNAS iterates over T devices to compute $g_{\Phi}^*$ and $g_{w}^*$, Figure 19 in Appendix J.2 shows that MODNAS generalizes well on 17 test devices with only 2 search devices due to its meta-learning capabilities. See Tables 5 and 3 in the Appendix for more details.

# J. Additional Experiments

## J.1. Predicted v/s Ground-Truth Latencies

In Figure 8, we present the scatter plots of the predictions of our hardware-aware `MetaPredictor` vs. the ground-truth latencies of different architectures. In the figure title we also report the kendall-tau correlation coefficient for every device. As observed, our predictor achieves high kendall-$\tau$ correlation coefficient across all devices.

## J.2. Additional Results on NAS-Bench-201

In Figure 13, we present the Pareto fronts obtained by our method in comparison to different baselines on the NAS-Bench-201 search space. In Figure 14, we present different additional metrics, such as GD and IGD (see Section E), to evaluate the quality of the Pareto fronts obtained on NAS-Bench-201. Figure 15 presents the Pareto front MODNAS yields when applying different latency constraints during the search phase. Figure 11a compares our method using the ReinMax gradient estimator to the GDAS estimator (Dong & Yang, 2019). As we can see, ReinMax obtains a qualitatively better hypervolume coverage compared to GDAS. Figure 12 presents the 3D Pareto front and hypervolume obtained by MODNAS compared to other baselines when optimizing for accuracy, latency and energy usage on NAS-Bench-201. Figure 18 presents the comparison of MODNAS with MGD to other gradient aggregation schemes, such as mean, sequential and MC sampling (see Section 4.1), across multiple hardware devices. Finally, in Figure 19 we present the robustness of MODNAS to the fraction of devices used for the predictor training and the search phase.

## J.3. Additional Results on Hardware-aware Transformers (En-De)

We show the Pareto fronts of MODNAS compared to baselines for the Transformer space in Figure 20, as well as their comparison with respect to hypervolume for the SacreBLEU metric in Figure 22. These results demonstrate the superior performance of our method compared to the other baselines on this benchmark. All evaluations are done by inheriting the weights of a pretrained supernet.

## J.4. Additional Results on the HW-GPT space

In figure 23, we present the Pareto fronts on all the 8 GPU types for MODNAS and different baselines. The Pareto fronts are obtained using the perplexity and energy predictors trained on data collected in the HW-GPT-Bench (Sukthanker et al., 2024).

## J.5. Additional Results on MobileNetV3

In Figure 24, we present the Pareto fronts of our method compared to different baselines for 12 different hardware devices on the MobileNetV3 space. We show as well the Pareto front of OFA+HELP (Lee et al., 2021b), ran with the original setting.

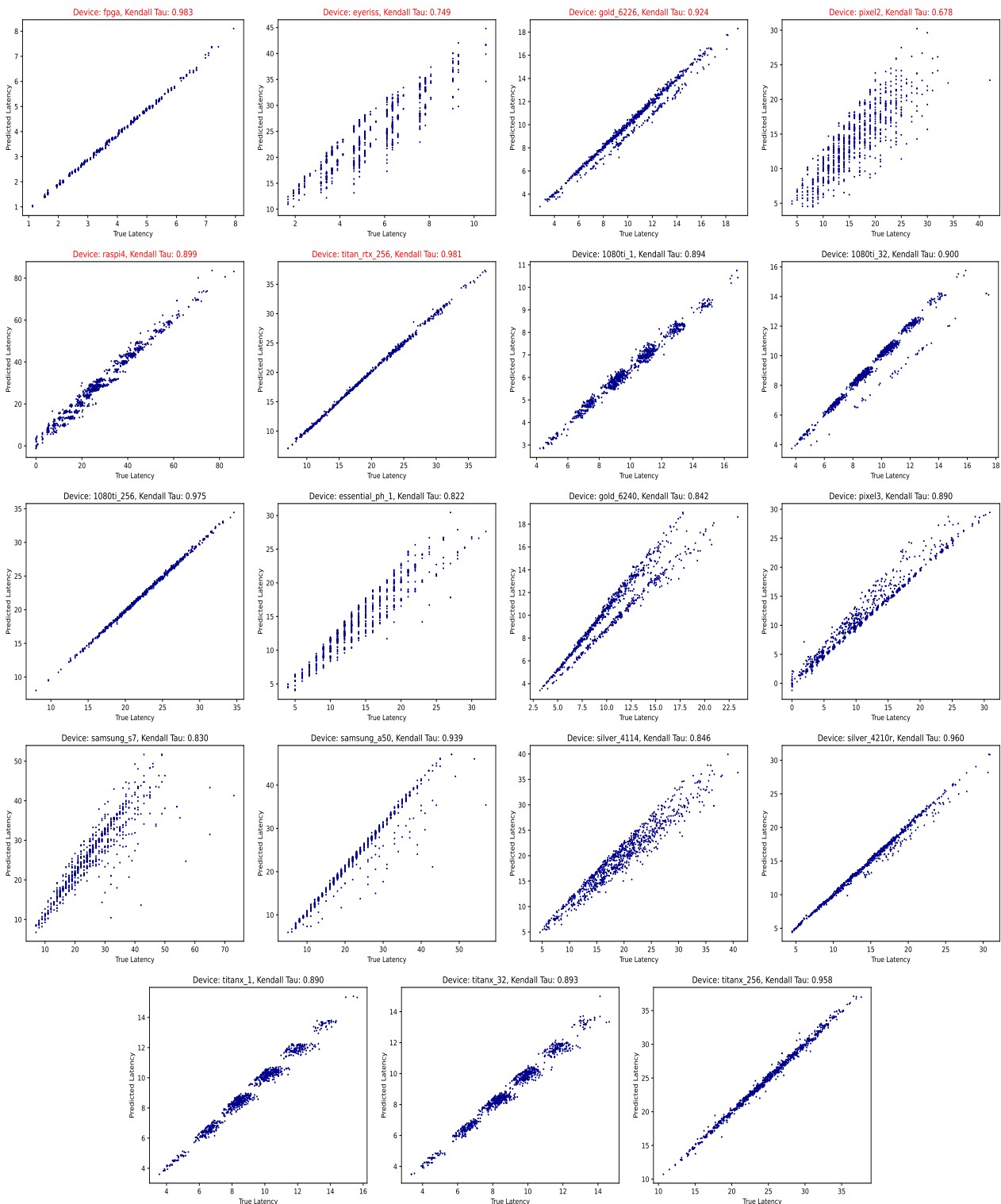

*Figure 10.* Scatter plots of predicted latencies from our pretrained `MetaPredictor` vs. ground-truth latencies (test devices in red).

*Table 4.* Hyperparameters used on different search spaces.

| Search Space | Hyperparameter Type | | Value |
|---|---|---|---|
| **NAS-Bench-201** | MetaHypernetwork | learning rate | 3e-4 |
| | | weight decay | 1e-3 |
| | | embedding layer size | 100 |
| | | hypernetwork bank size | 50 |
| | | optimizer | Adam |
| | | ReinMax temperature | 1 |
| | Supernetwork | learning rate | 0.025 |
| | | momentum | 0.9 |
| | | weight decay | 0.0027 |
| | | learning rate scheduler | cosine |
| | | epochs | 100 |
| | | batch size | 256 |
| | | gradient clipping | 5 |
| | | cutout | true |
| | | cutout length | 16 |
| | | initial channels | 16 |
| | | optimizer | SGD |
| | | train portion | 0.5 |
| **MobileNetV3 (OFA)** | MetaHypernetwork | learning rate | 1e-5 |
| | | weight decay | 1e-3 |
| | | embedding layer size | 100 |
| | | hypernetwork bank size | 50 |
| | | optimizer | Adam |
| | | ReinMax temperature | 1 |
| | Supernetwork | learning rate | 1e-3 |
| | | momentum | 0.9 |
| | | weight decay | 3e-5 |
| | | learning rate scheduler | cosine |
| | | epochs | 50 |
| | | batch size | 32 |
| | | bn_momentum | 0.1 |
| | | bn_eps | 1e-5 |
| | | dropout | 0.1 |
| | | width | 1.2 |
| | | optimizer | SGD |
| | | train portion | 1.0 |
| **Seq-Seq Transformer (HAT)** | MetaHypernetwork | learning rate | 3e-4 |
| | | weight decay | 1e-3 |
| | | embedding layer size | 100 |
| | | hypernetwork bank size | 50 |
| | | optimizer | Adam |
| | | ReinMax temperature | 1 |
| | Supernetwork | learning rate | 1e-7 |
| | | momentum | 0.9 |
| | | weight decay | 0.0 |
| | | learning rate scheduler | cosine |
| | | epochs | 110 |
| | | batch size/max-tokens | 4096 |
| | | criterion | label_smoothed_cross_entropy |
| | | attention-dropout | 0.1 |
| | | dropout | 0.3 |
| | | precision | float32 |
| | | optimizer | Adam |
| | | train portion | 1.0 |
| **HW-GPT-Bench** | MetaHypernetwork | learning rate | 1e-5 |
| | | weight decay | 1e-3 |
| | | embedding layer size | 100 |
| | | hypernetwork bank size | 50 |
| | | optimizer | Adam |
| | | ReinMax temperature | 1 |
| | Supernetwork | learning rate | 0.000316 |
| | | momentum | - |
| | | weight decay | 0.1 |
| | | learning rate scheduler | cosine |
| | | steps | 800k |
| | | batch size/max-tokens | 32768 |
| | | criterion | cross_entropy |
| | | attention-dropout | 0.0 |
| | | dropout | 0.0 |
| | | precision | bfloat16 |
| | | optimizer | AdamW |
| | | train portion | 1.0 |

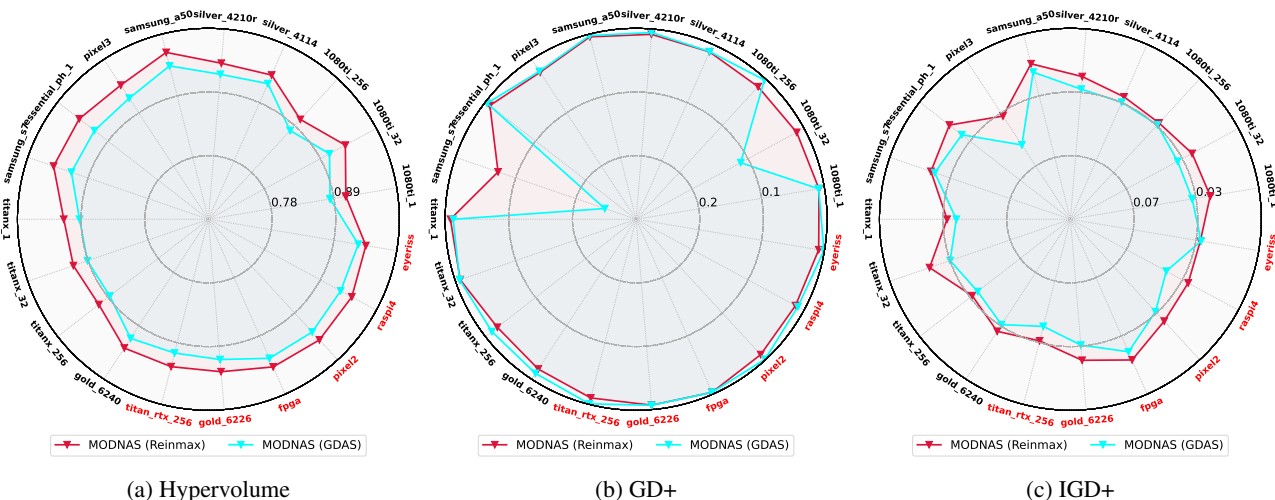

(a) Hypervolume      (b) GD+      (c) IGD+

*Figure 11.* Hypervolume, GD+ and IGD+ of MODNAS with Reinmax as gradient estimator in the `Architect` vs. the one from GDAS (Dong & Yang, 2019) across 19 devices on NAS-Bench-201. Higher area in the radar indicates better performance for every metric. Test devices are colored in red around the radar plot.

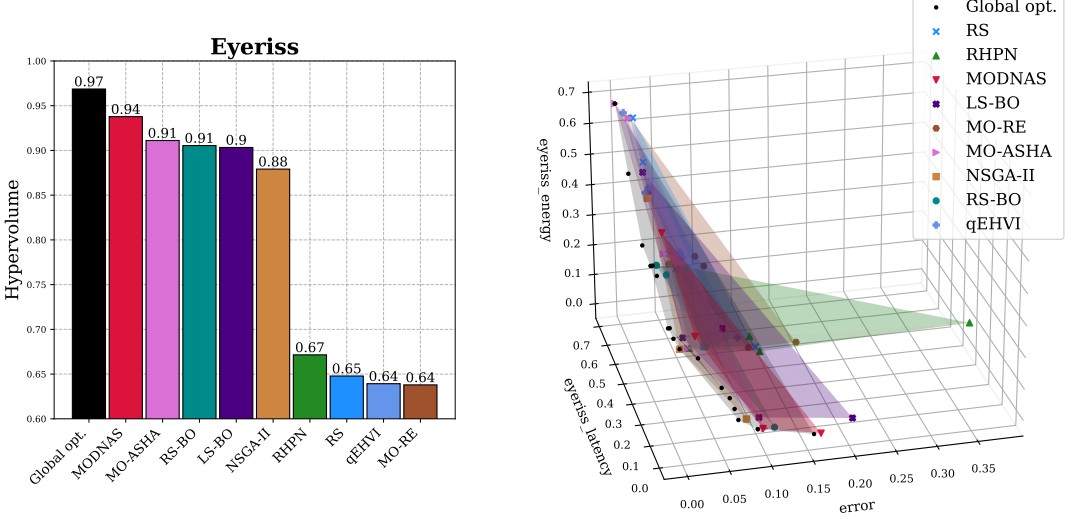

*Figure 12.* HV (*left*) and Pareto front (*right*) of MODNAS and baselines on Eyeriss with 3 normalized objectives: error, latency and energy usage. HV was computed using the $(1, 1, 1)$ reference point on the right 3D plot.

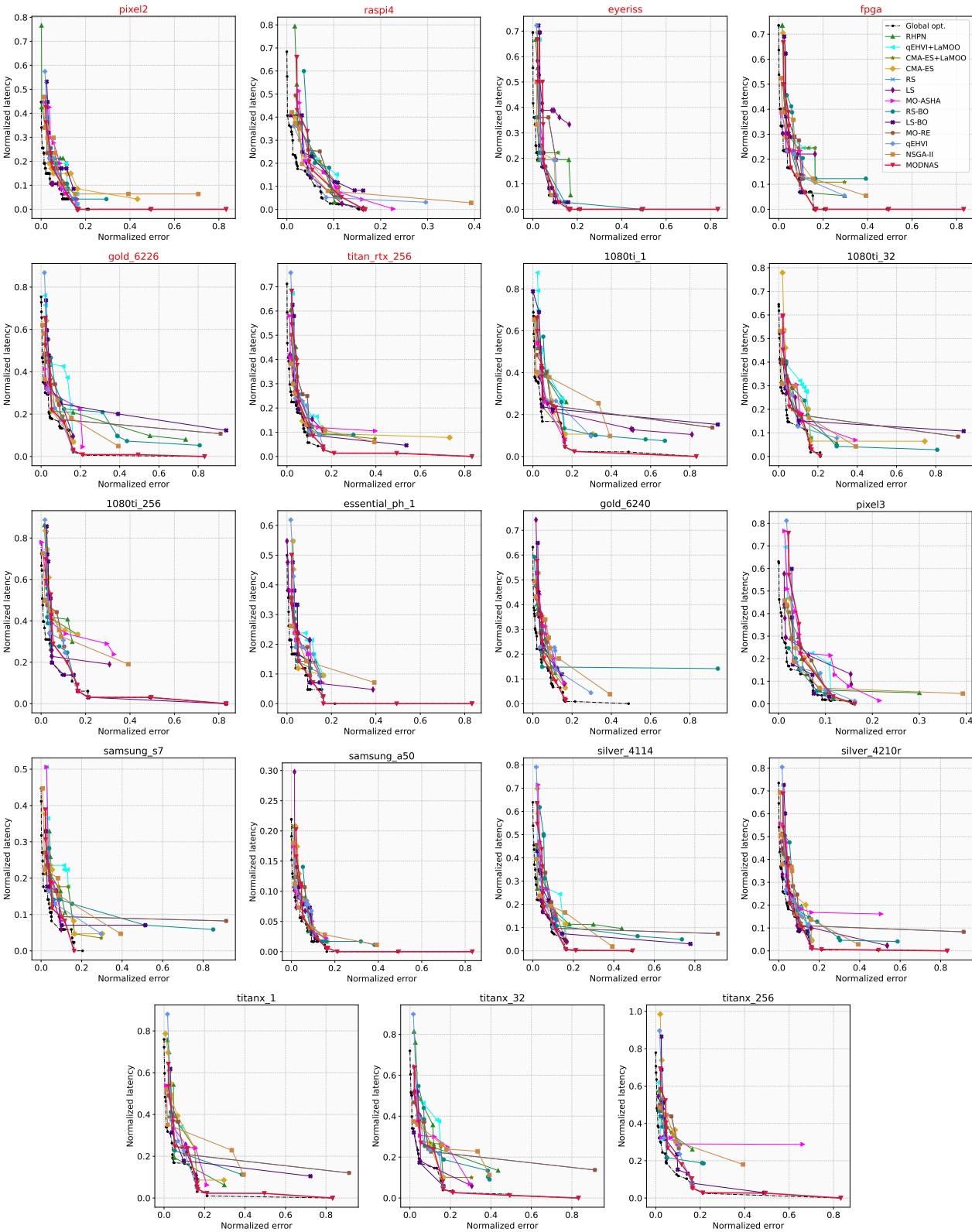

*Figure 13.* Pareto fronts of MODNAS and baselines on NAS-Bench-201. MODNAS-SoTL is not shown for better visibility.

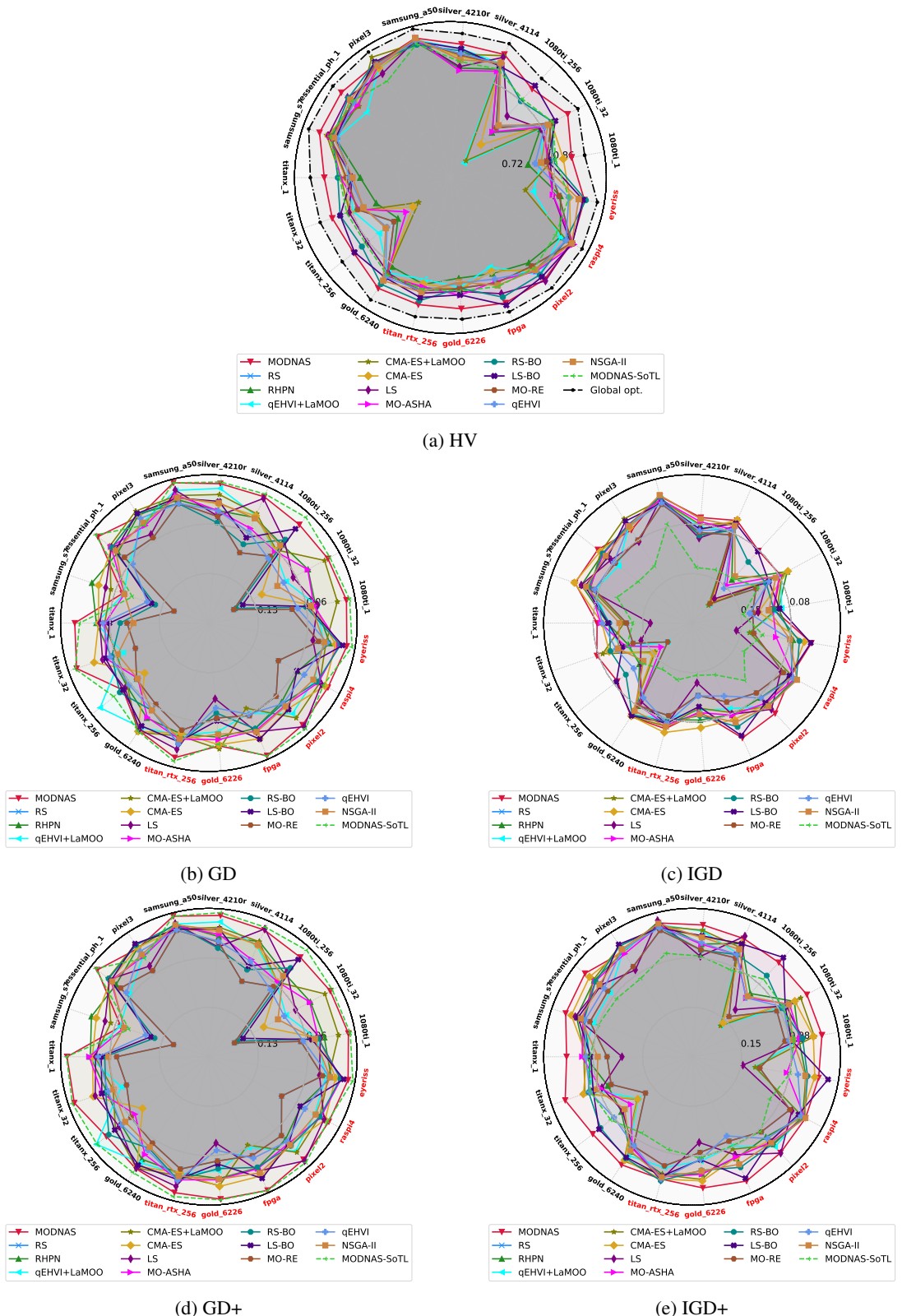

(a) HV

(b) GD

(c) IGD

(d) GD+

(e) IGD+

*Figure 14.* HV, GD, GD+, IGD and IGD+ of MODNAS and baselines across 19 devices on NAS-Bench-201. For every device we optimize for 2 objectives, namely *latency (ms)* and *test accuracy* on CIFAR-10. For method, metric and device we report the mean of 3 independent search runs. Higher area in the radar indicates better performance for every metric. Test devices are colored in red around the radar plot. Here we allocate double the budget to baselines, i.e. we run all baselines for 50 function evaluations.

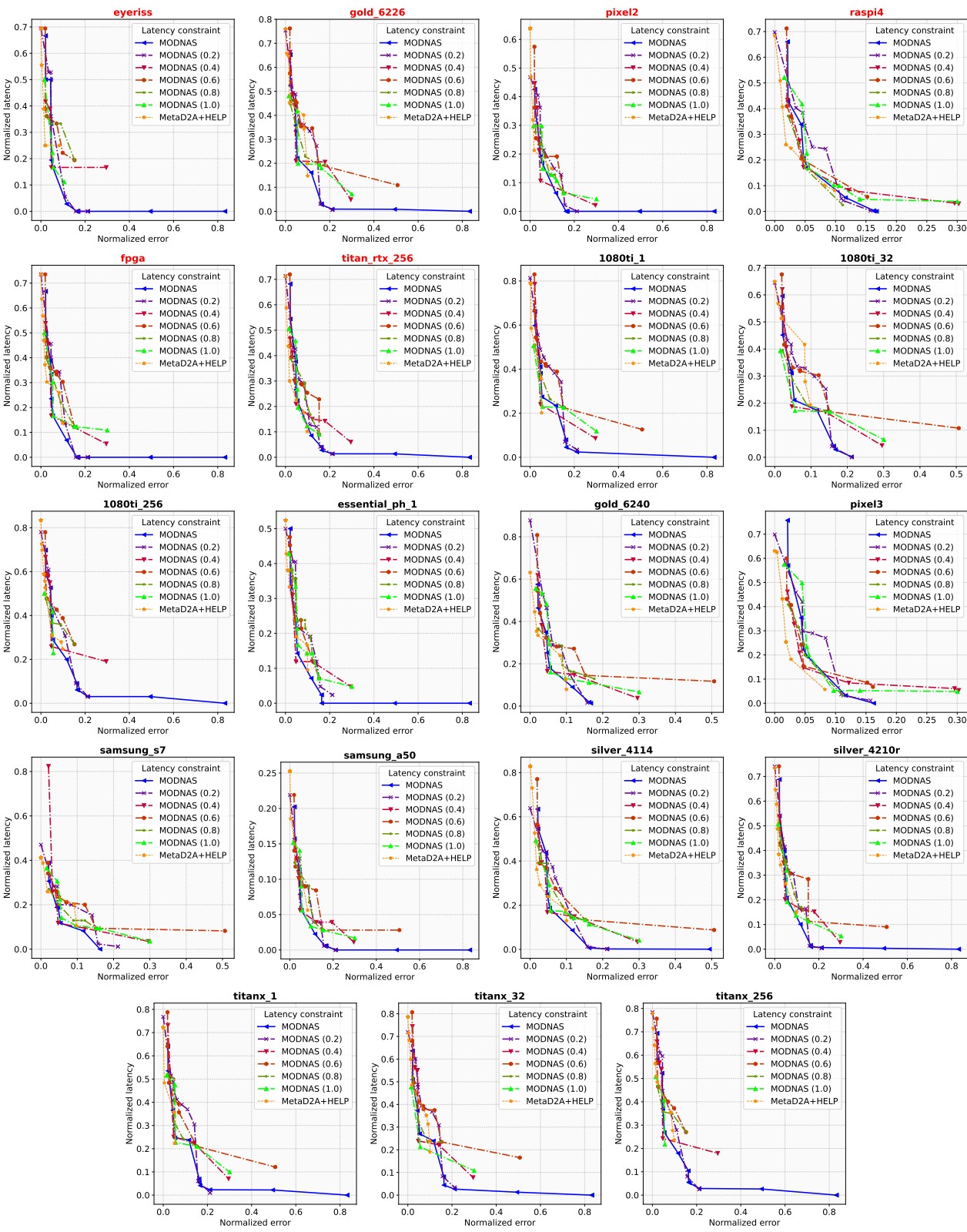

*Figure 15.* Pareto fronts of MODNAS ran with different latency constraints during search.

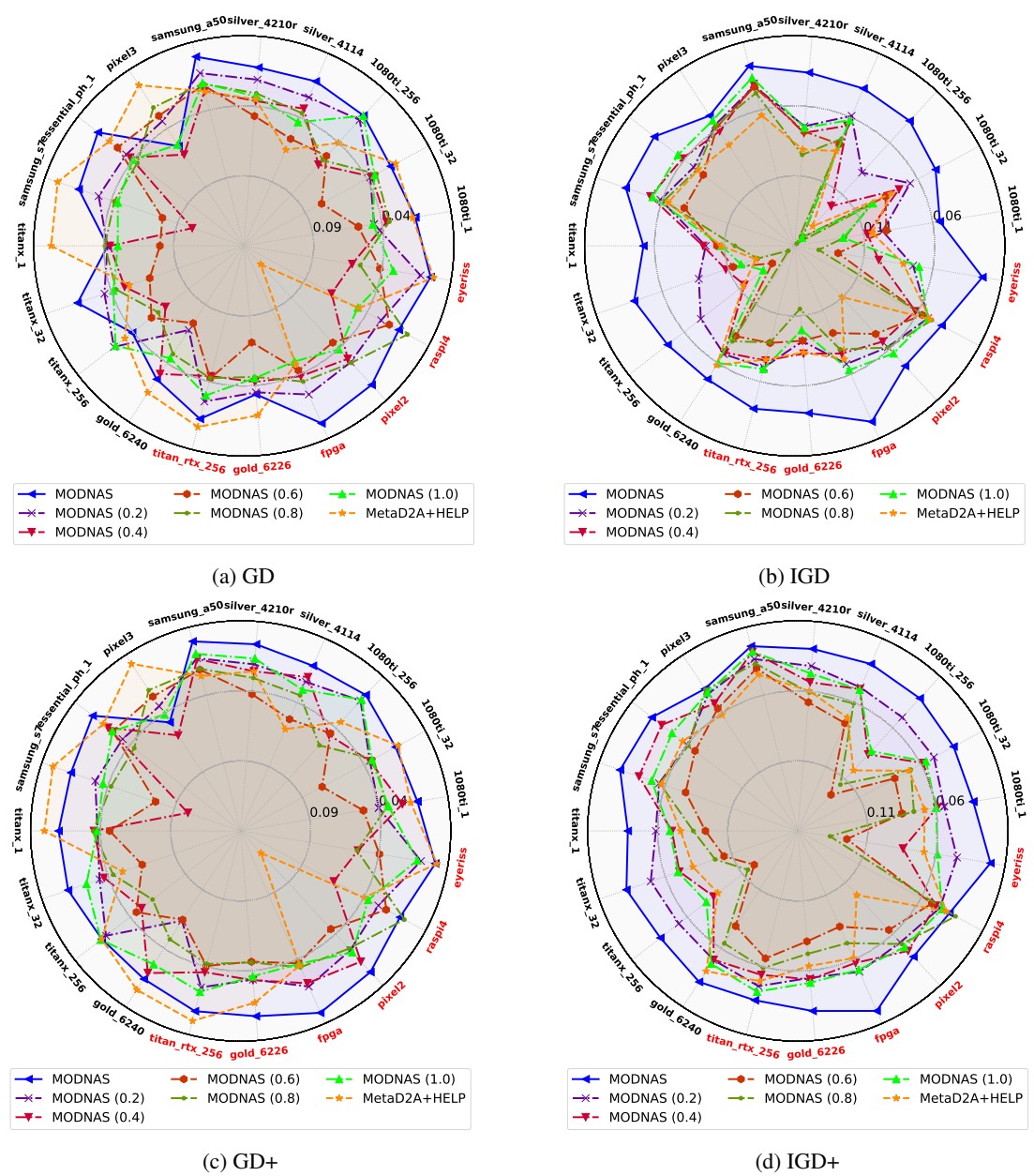

*Figure 16.* GD, GD+, IGD and IGD+ of MODNAS with different latency constraints during search across 19 devices on NAS-Bench-201. Higher area in the radar indicates better performance for every metric. Test devices are colored in red around the radar plot.

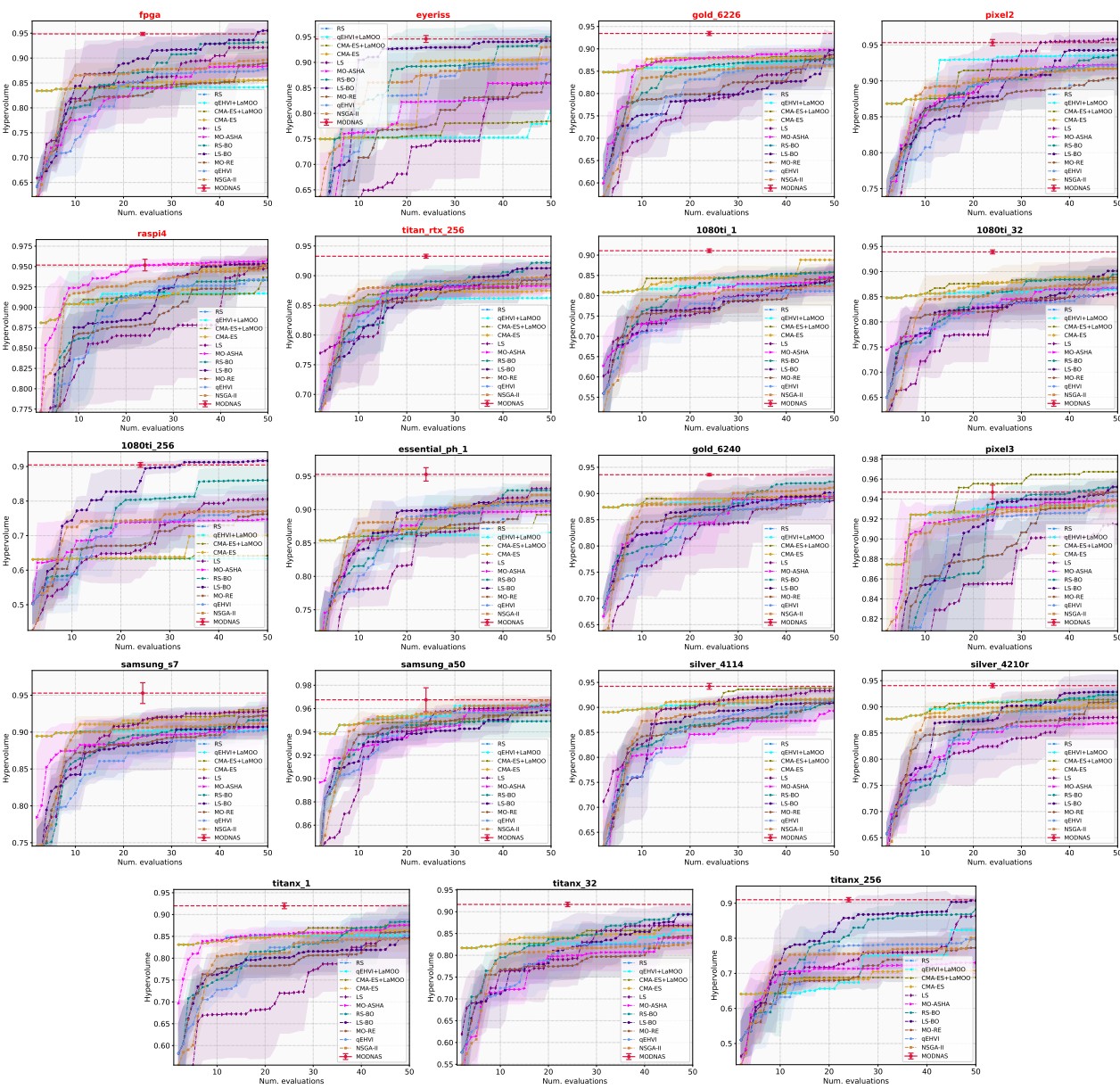

*Figure 17.* HV over number of evaluated architectures on NAS-Bench-201 of MODNAS and the blackbox MOO baselines. Note that for MODNAS we only have 24 evaluations in the end.

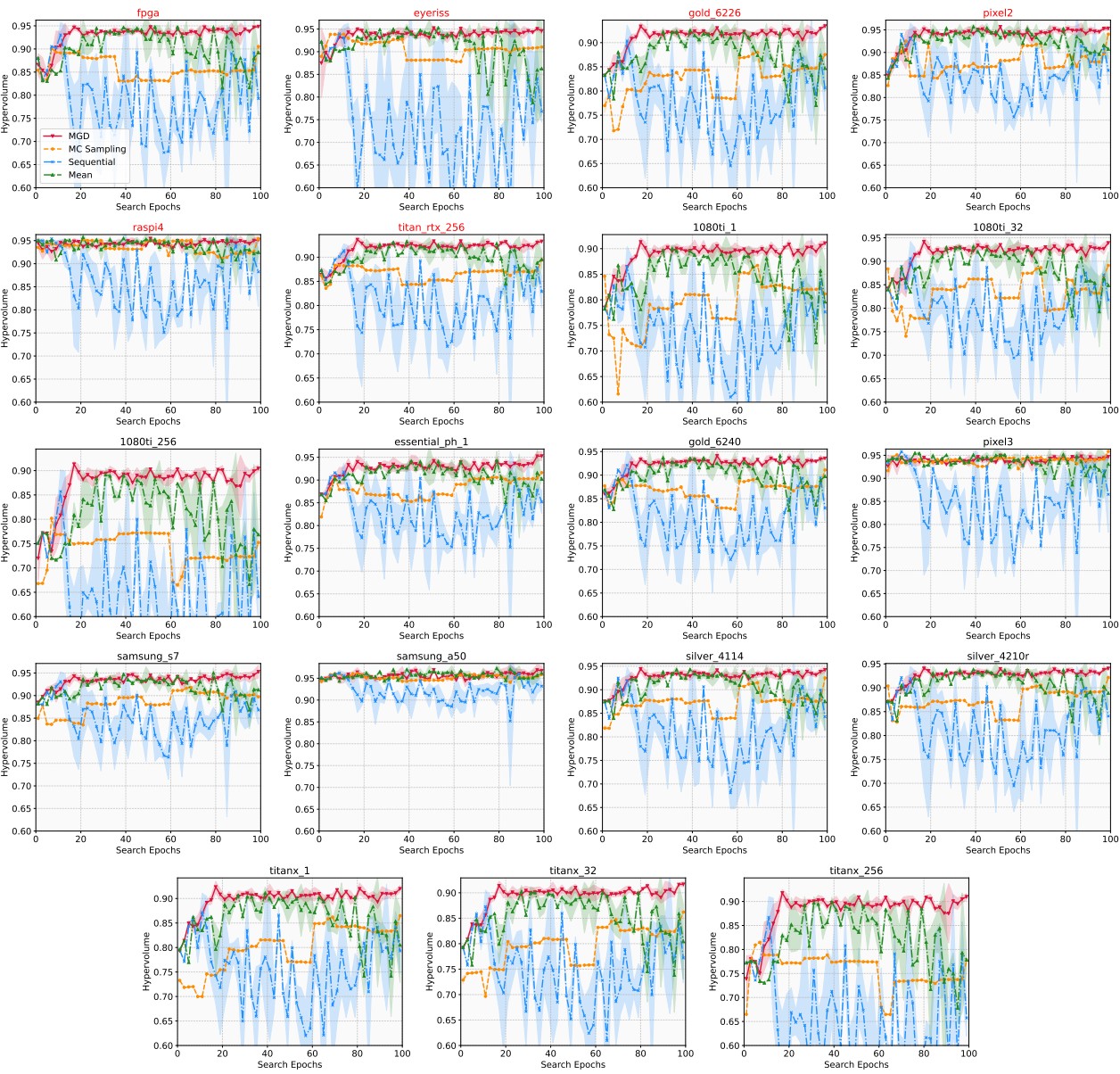

*Figure 18.* HV over time on NAS-Bench-201 of MODNAS with different gradient update schemes.

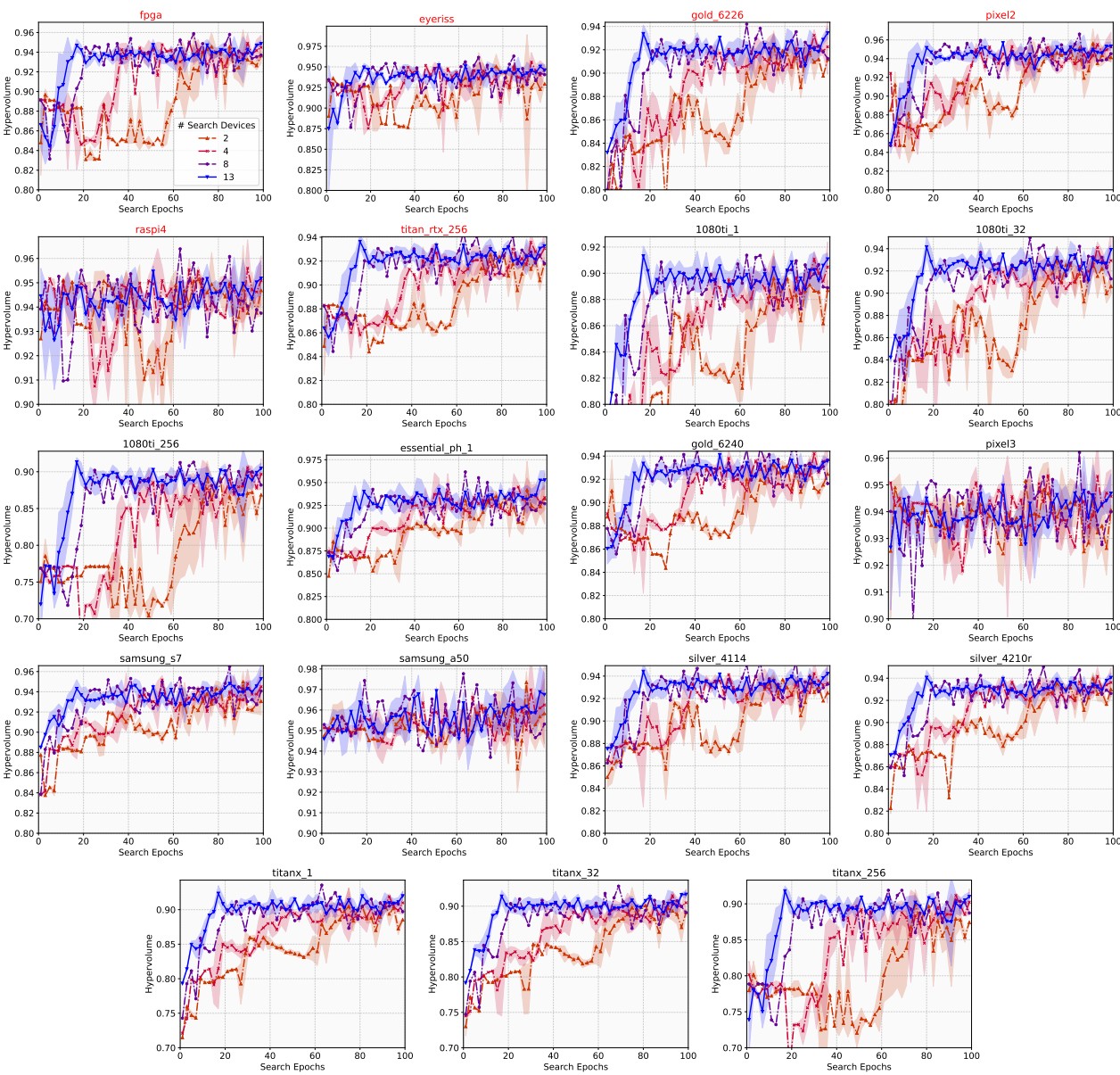

*Figure 19.* HV over time on NAS-Bench-201 of MODNAS with different number of devices during search. For number of devices less than 13 (default one) we randomly select a subset from these 13 devices.

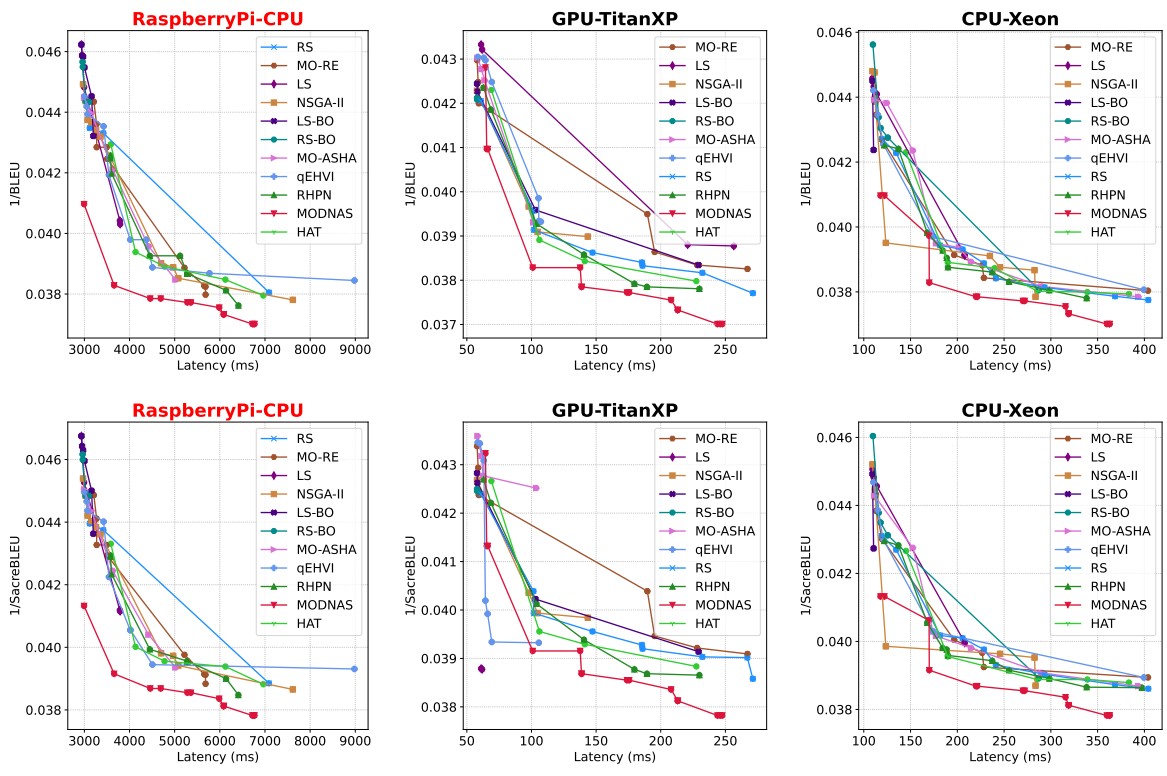

*Figure 20.* Pareto fronts of MODNAS and baselines on the HAT space for the WMT' En-De task. All performance metrics are obtained from the inherited supernet weights.

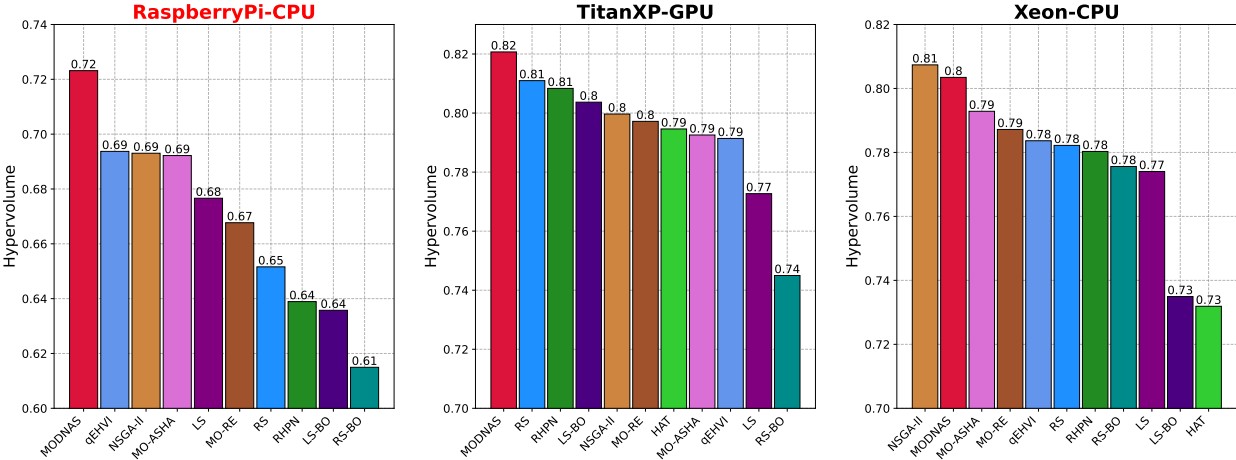

*Figure 21.* Hypervolume (HV) of MODNAS and baselines across devices on the HAT space. The objectives used to compute the HV are latency and BLEU score. Leftmost plot is for the test device.

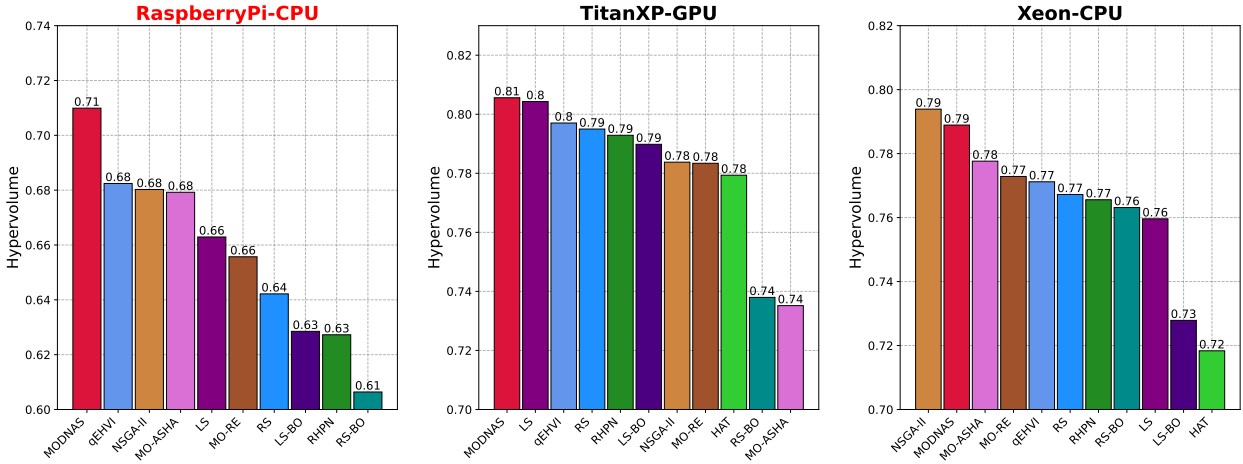

*Figure 22.* Hypervolume (HV) of MODNAS and baselines across devices on the HAT space. The objectives used to compute the HV are latency and SacreBLEU score. Leftmost plot is for the test device. MODNAS is the best or on par to the baselines across all three devices.

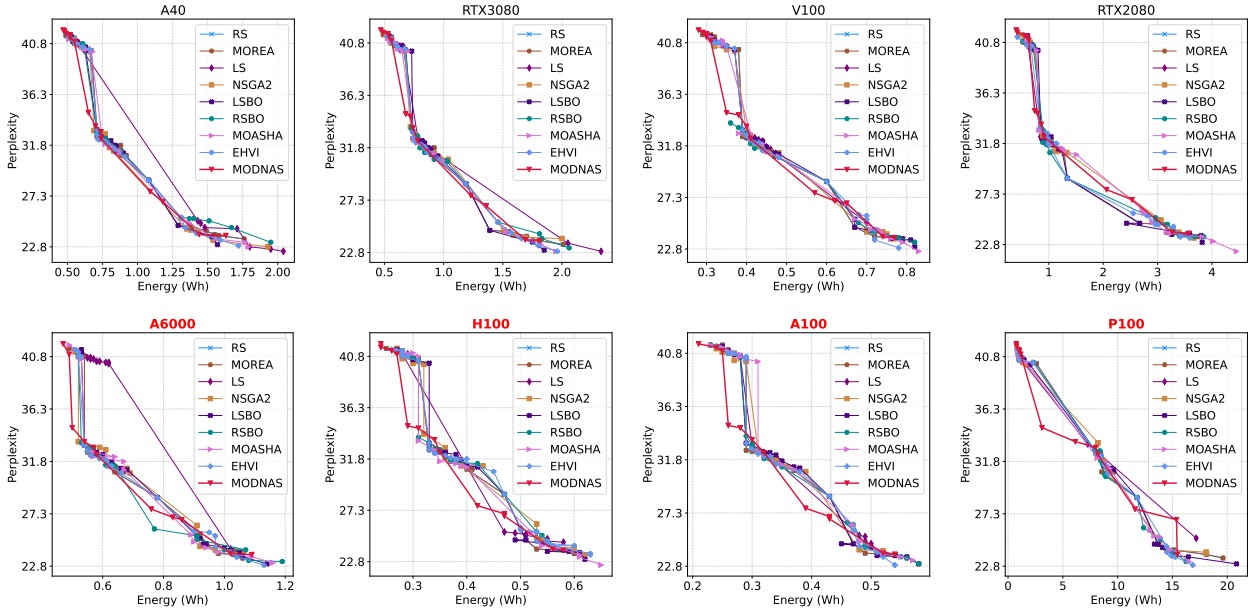

*Figure 23.* Pareto fronts of MODNAS and baselines optimizing for GPU energy consumption (Wh) and perplexity on the HW-GPT-Bench space.

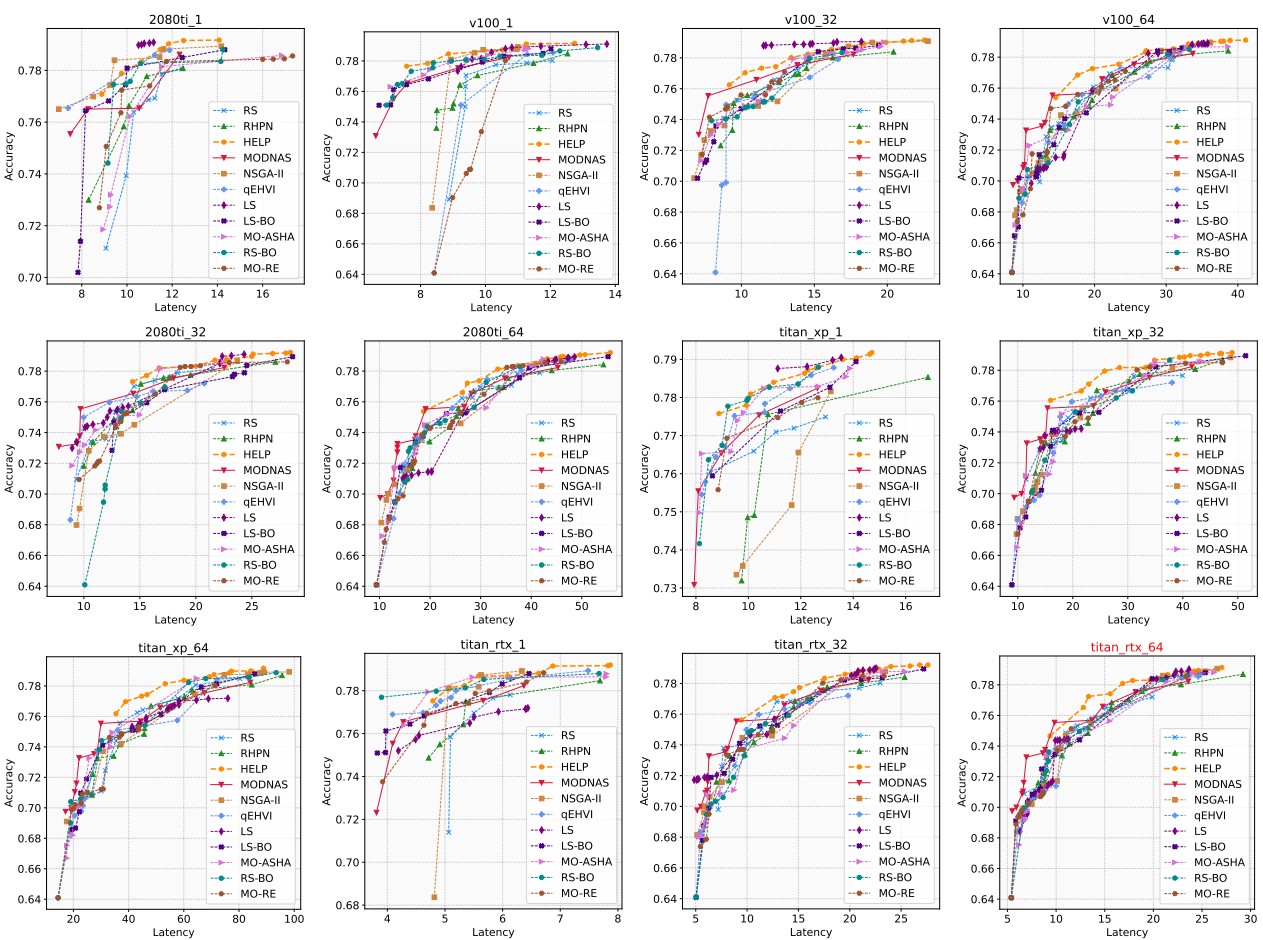

*Figure 24.* Pareto fronts of MODNAS and baselines on the MobileNetV3 space.

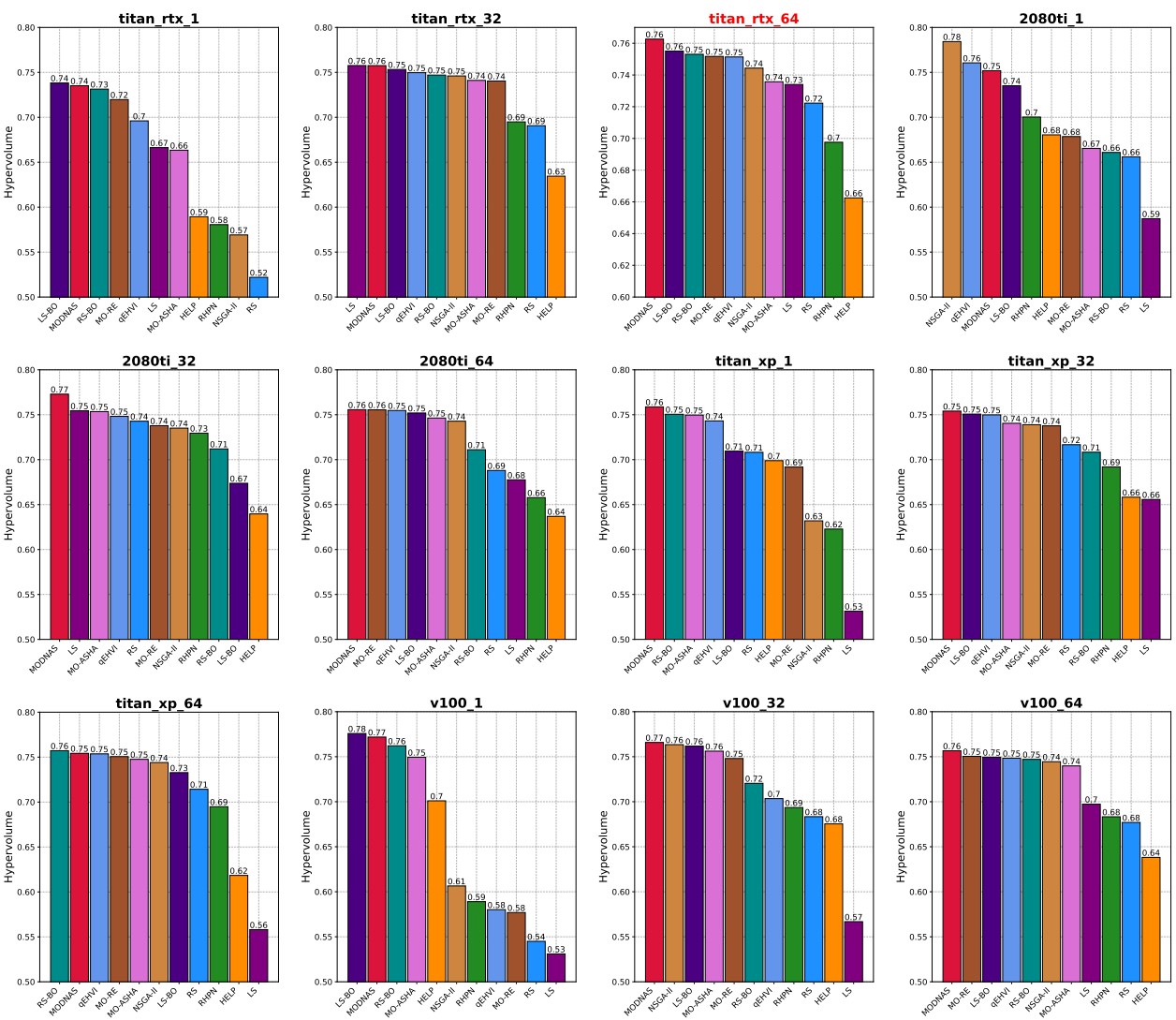

*Figure 25.* Hypervolume across devices on the MobileNetV3 search space of MODNAS and baselines. Here the Nvidia Titan RTX is the test device.

