# OpenReview forum: "Multi-objective Differentiable Neural Architecture Search"
_ICML.cc/2024/Workshop/WANT — WANT@ICML 2024 Poster_

### Official Review · Reviewer_xdDy · 2024-06-13
**A multi-model algorithms for neural architecture search balancing accuracy performances and hardware requirements**

**Confidence:** 2

**Summary:**

The authors propose a multi-objective differentiable neural architecture search algorithm (MODNAS). The algorithm allows to predict the Pareto front for multi-objective optimization, where the objectives include model accuracy and hardware latency. The algorithm relies on several models. First, an _MetaHyperNetwork_ predicts an unnormalized architecture distribution based on hardware latency inputs and objective preferences. Then, from this distribution an _Architect_ sampler that returns discrete architectural configuration. A _Supernetwork_ estimates the accuracy of the selected architecture, while _MetaPredictor_ estimates hardware objectives. The _MetaHyperNetwork_ is trained with multi-objective gradient descent. At inference, the Pareto front is provided by generating predictions for 24 different preference vectors.

**Strengths:**

- The proposed architecture has been tested on 4 different search spaces and with 19 different hardware;
- The paper provide an extensive set of experiments and comparisons with baseline;
- Compared on the hypervolume of the Pareto front, the propose approach outperforms several baselines from the literature.

**Weaknesses:**

- The paper is dense (and long), with many key concepts and algorithms. It would gain in clarity by providing a more organized and progressive description;
- The training process and especially the data collection could be detailed more explicitly;
- Although the authors compare experimentally MODNAS to many other approaches, it would have been interesting to emphasize on the core theoretical differences. (I might also sensitive to this point because of my lack of expertise in NAS);
- Although MODNAS could theoretically accommodate more hardware features, it would be interesting to check results on more features closer to real applications.

---

### Meta-Review · Area_Chair_xuaD · 2024-06-17

**Recommendation:** Accept (Poster)
**Confidence:** 4

**Metareview:**

The paper received a single review. Upon checking the paper, the AC is positive about the proposed method, MODNAS, a new approach for hardware-aware NAS that encodes user preferences for the trade-off between performance and hardware metrics, and yields representative and diverse architectures in just one run. The method is tested on a large number of hardware targets and benchmarks, where it shows promising results.

---

### Decision · Program_Chairs · 2024-06-17

**Decision:**

Accept (Poster)

**Comment:**

We thank the authors for their time and contribution to WANT and we are pleased to share that after the reviewing process the paper has been accepted. Congratulations! We encourage the authors to consider reviewers' feedback for the improvement of the camera-ready version. We hope to see you in person at the workshop and brainstorm on efficient training research together!